# Peli1 negatively regulates noncanonical NF-κB signaling to restrain systemic lupus erythematosus

Junli Liu[1], Xinfang Huang[2], Shumeng Hao[1], Yan Wang[1], Manman Liu[2], Jing Xu[1], Xingli Zhang[1], Tao Yu[1], Shucheng Gan[1], Dongfang Dai[3], Xuan Luo[3], Qingyan Lu[3], Chaoming Mao[3], Yanyun Zhang[1], Nan Shen[1,4], Bin Li[5], Mingzhu Huang[6], Xiaodong Zhu[6], Jin Jin[7], Xuhong Cheng[8], Shao-Cong Sun[8] & Yichuan Xiao [1]

Systemic lupus erythematosus (SLE) is characterized by uncontrolled secretion of auto-antibodies by plasma cells. Although the functional importance of plasma cells and auto-antibodies in SLE has been well established, the underlying molecular mechanisms of controlling autoantibody production remain poorly understood. Here we show that Peli1 has a B cell-intrinsic function to protect against lupus-like autoimmunity in mice. Peli1 deficiency in B cells induces autoantibody production via noncanonical NF-κB signaling. Mechanically, Peli1 functions as an E3 ligase to associate with NF-κB inducing kinase (NIK) and mediates NIK Lys48 ubiquitination and degradation. Overexpression of Peli1 inhibits noncanonical NF-κB activation and alleviates lupus-like disease. In humans, PELI1 levels negatively correlate with disease severity in SLE patients. Our findings establish Peli1 as a negative regulator of the noncanonical NF-κB pathway in the context of restraining the pathogenesis of lupus-like disease.

---

[1] The Key Laboratory of Stem Cell Biology, Institute of Health Sciences, Shanghai Institutes for Biological Sciences, Chinese Academy of Sciences, University of Chinese Academy of Sciences, 200031 Shanghai, China. [2] Department of Nephrology and Rheumatology, Xin Hua Hospital Affiliated to Shanghai Jiao Tong University School of Medicine, 200092 Shanghai, China. [3] Department of Nuclear Medicine, The Affiliated Hospital of Jiangsu University, 438 Jiefang Road, 212001 Zhenjiang, China. [4] Shanghai Institute of Rheumatology, Shanghai Renji Hospital, Shanghai Jiao Tong University School of Medicine, 200001 Shanghai, China. [5] Shanghai Institute of Immunology, Shanghai Jiao Tong University School of Medicine, 200025 Shanghai, China. [6] Department of Medical Oncology, Fudan University Shanghai Cancer Center, Shanghai, China. [7] Life Sciences Institute, Zhejiang University, 310058 Hangzhou, China. [8] Department of Immunology, MD Anderson Cancer Center, The University of Texas, Houston, TX 77030, USA. Correspondence and requests for materials should be addressed to Y.X. (email: ycxiao@sibs.ac.cn)

Systemic lupus erythematosus (SLE) is a complex, multi-system autoimmune disease with the etiology of a combination of genetic and environmental factors. The hallmark of SLE is an uncontrolled B cell production of autoantibodies specific for nuclear antigens such as double-stranded DNA (dsDNA) and chromatin etc., resulting in the formation and deposition of immune complexes to cause tissue damage[1–3]. The mature B cells are activated when encountering with antigens, which induce B cell proliferation and the immunoglobulin class switching, finally exhibit specific function through secreted diversified antibodies[4]. Accumulating evidences from experimental and clinical data indicate that B cells are essential for the pathogenesis of SLE[5–8]. In addition, deletion of B cells or inhibiting B cell activation has been applied for clinically approved therapeutic strategies during SLE treatment[9–13].

It is known that noncanonical NF-κB signaling that induced by CD40 ligand (CD40L), B cell-activating factor (BAFF), etc., is critical for the antibody production in activated B cells[14,15]. Previous studies have demonstrated that the activation of noncanonical NF-κB pathway by these inducers is dependent on the NF-κB inducing kinase (NIK), which activate IKKα to induce p100 processing to p52, causing the translocation of p52/RelB heterodimer into nucleus[16,17]. Accordingly, either NIK inactivation or functional mutation of p100 impairs the antibody secretion and B cell-mediated immune responses[18,19]. In contrast, mice overexpressing BAFF (BAFF-Tg mice) exhibit hyperactivation of noncanonical pathway and develop an autoimmune lupus-like disease with increasing production of autoantibodies[20–22].

The activation of noncanonical NF-κB pathway depends on the accumulation of NIK[14,15], which is tightly regulated by the ubiquitination system. Under homeostasis, TRAF3 links NIK to TRAF2-cIAPs E3 complex, thereby promoting cIAPs-mediated Lys48-linked NIK polyubiquitination and degradation[23,24]. Thus, activation of noncanonical NF-κB involves signal-induced regulation of NIK ubiquitination, but how this event is regulated is not fully understood. The Peli (also called Pellino) family of proteins are a type of E3 ubiquitin ligases, and mediate the formation of both Lys63- or Lys48-linked polyubiquitin chains. We and others have demonstrated that Peli1 is critical for the regulation of toll-like receptor (TLR) and interleukin-1 receptor (IL-1R) signaling in innate immune cells[25–27], and modulates T cell receptor (TCR) signaling in T cells[28]. Our study suggested that Peli1 controls TLR-mediated TRAF3 degradation and MAPK activation, leading to microglia activation and autoimmune inflammation in central nervous system[29].

In the present study, we uncover a crucial role for Peli1 in B cell autoantibody production and SLE pathogenesis. We also provide molecular and genetic evidence that Peli1 serves as an E3 ubiquitin ligase of NIK, regulating Lys48-linked ubiquitination of NIK and noncanonical NF-κB activation.

## Results

**Peli1 deficiency promotes B cell activation.** We previously found that *Peli1* is highly expressed in mouse splenic B cells[28] and in human CD19+ B cells (BioGPS data), but whether and how Peli1 may affect B cell function and SLE pathogenesis is still

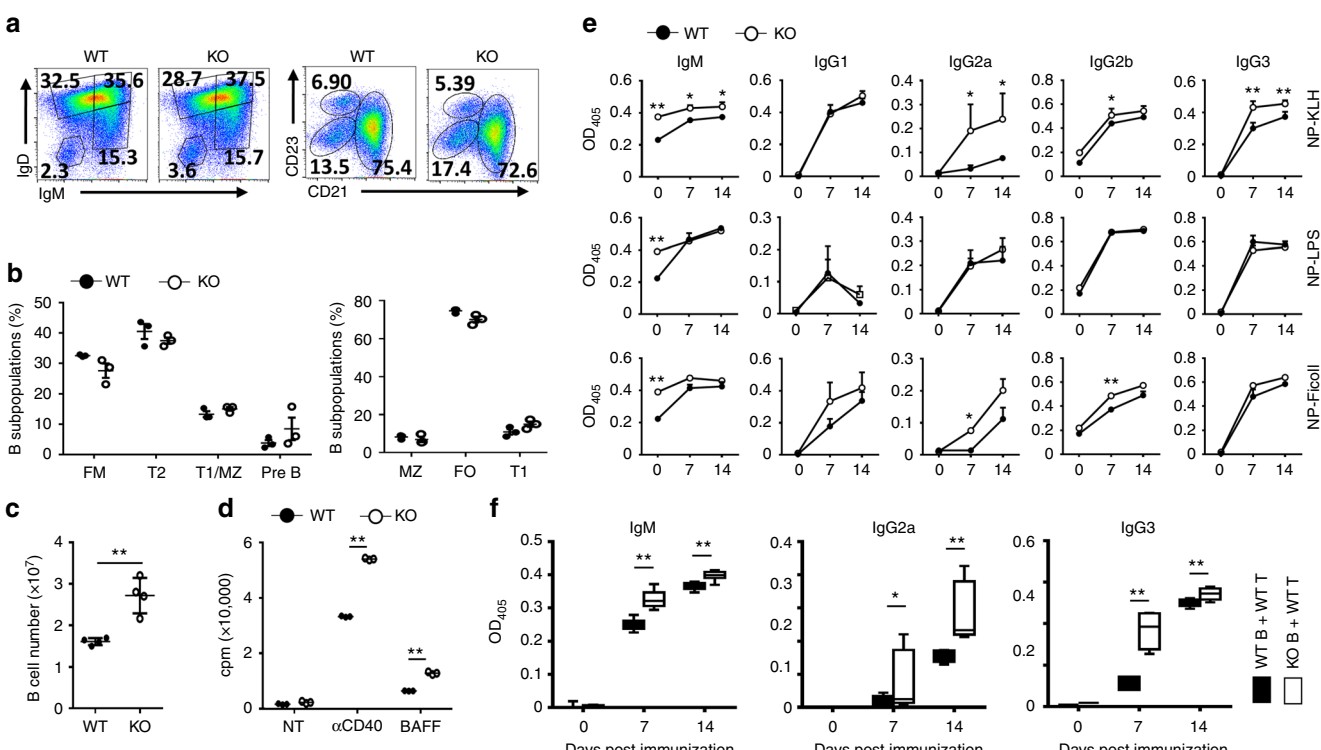

**Fig. 1** *Peli1* deficiency promotes B cell proliferation and antibody secretion. **a**, **b** Flow cytometric analysis of the percentages of B cell subpopulations in the spleens of WT and *Peli1*-dificient mice. Data are presented as a representative plot (**a**) and summary graph (**b**). **c** The absolute numbers of B220+ B cells in the spleens of WT and KO mice. **d** Proliferation of WT and KO splenic B cells incubated in vitro for 72 h in the absence (NT) or presence of anti-CD40 (αCD40) or BAFF, then assessed by [3H]thymidine incorporation. **e** Enzyme-linked immunosorbent assay (ELISA) of NP-specific antibody isotypes in the serum of WT and *Peli1*-dificient mice immunized intraperitoneally with NP-KLH, NP-LPS, or NP-Ficoll. **f** ELISA of NP-specific IgM, IgG2a and IgG3 in the serum of Rag1-dificient mice that transferred with WT T cells plus WT or KO B cells, and then immunized intraperitoneally with NP-KLH. Data are shown as the mean ± SEM based on three independent experiments. Two-tailed Student's *t*-tests were performed. *P < 0.05 and **P < 0.01

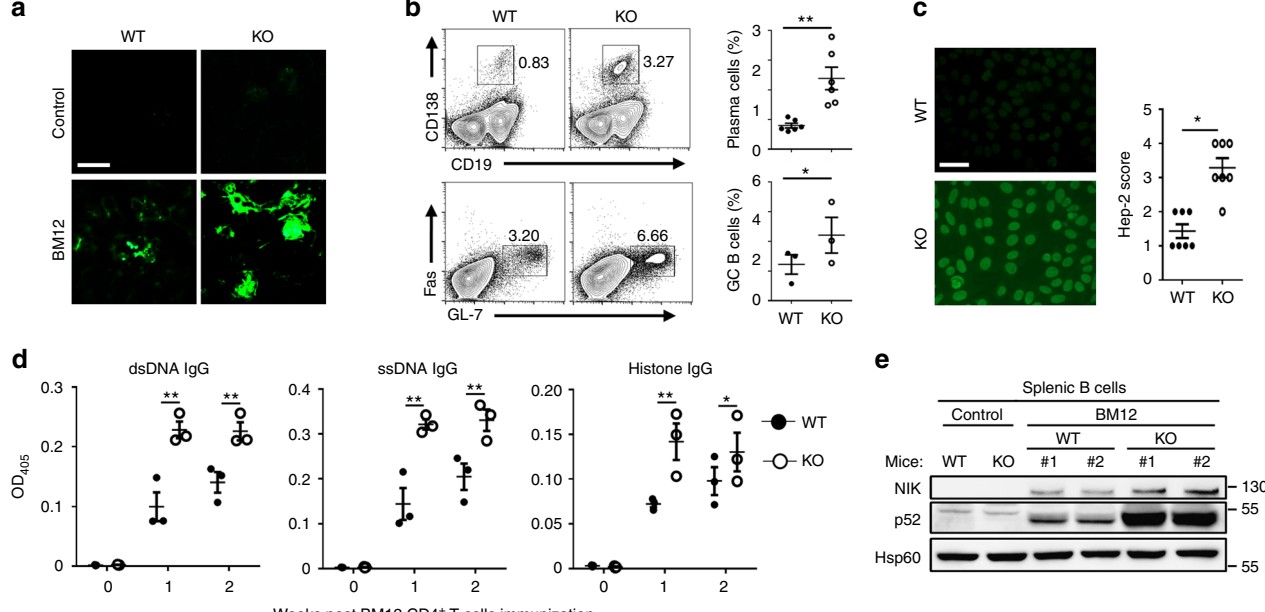

**Fig. 2** Peli1 deficiency aggravates the induction of lupus-like disease. **a** WT and *Peli1*-deficient (KO) mice were intraperitoneally injected with 7.5 million of CD4$^+$ T cell from C57BL/6 mice (control) or from BM12 mice. Representative immunofluorescent images showing IgG deposits in kidney by staining with Alexa Fluor 488-labeled anti-mouse IgG. Scale bar, 100 μm. **b** Flow cytometric analysis of the percentages of CD19$^-$CD138$^+$ plasma cells and Fas$^+$GL-7$^+$ germinal center (GC) B cells in WT and KO immunized mice as described in (**a**). Data are presented as a representative plot (left panel) and summary graph (right panel). **c** Distinct anti-nuclear antibody (ANA) staining patterns of the Hep-2 cell line with serum from WT and KO mice 4 weeks after immunization as described in (**a**). Scale bar, 50 μm. **d** Enzyme-linked immunosorbent assay (ELISA) of anti-dsDNA, anti-ssDNA, and anti-histone IgG in serum from WT and KO immunized mice as described in (**a**) at the indicate time point. **e** Immunoblot showing NIK and p52 protein levels in splenic B cells from control and BM12 CD4$^+$ T cells immunized WT and KO mice as described in (**a**). Data are shown as the mean ± SEM based on three independent experiments. Two-tailed Student's *t*-tests were performed. *$P < 0.05$ and **$P < 0.01$

unknown. Taking advantage of *Peli1*-knockout (KO) mice and their wild-type (WT) littermate, we examined the role of Peli1 on B cells development, proliferation and antibody production ability. The results indicated that there are comparable frequencies of B cell sub-types both in spleens (Fig. 1a, b) and BMs (Supplementary Fig. 1a, 1b) between WT and KO mice. However, the absolute number of CD19$^+$ B cell in the spleen significantly increased in *Peli1*-deficienct mice ($P < 0.01$, by Student's *t*-tests, Fig. 1c). We previously reported and now confirmed that *Peli1* deficiency is dispensable for BCR-induced but impaired TLR-induced B cell proliferation[27] (Supplementary Fig. 1c), which promote us to speculate that the incensement of B cells in *Peli1*-deficient mice may attribute to the promotion of noncanonical NF-κB signaling. Indeed, the proliferation of KO B cells was significantly enhanced upon the stimulation by anti-CD40 and BAFF ($P < 0.01$, by Student's *t*-tests, Fig. 1d), two major non-canonical NF-κB inducers for B cells.

CD40L is a costimulatory molecule that is expressed on T cells, and mediates B cell activation and antibody production through its ligation with CD40, which promote the noncanonical NF-κB activation and represent the T cell-dependent stimulation for B cells[17]. To examine whether Peli1 mediates CD40L-induced antibody secretion in vivo, we immunized WT and KO mice with T cell-dependent antigen 4-hydroxy-3-nitrophenylacetyl (NP)-keyhole lympet hemocyanin (KLH), as well as with T cell-independent antigens, including NP-LPS and NP-Ficoll as controls. The data showed that Peli1 deficiency dramatically promoted the secretion of antigen-specific IgM, IgG2a, and IgG3 in the sera when the mice were immunized with NP-KLH ($P < 0.05$ and $P < 0.01$, by Student's *t*-tests, Fig. 1e). In contrast, deletion of Peli1 didn't affect the ability of B cells to produce antigen-specific antibodies in response to NP-LPS immunization,

and caused a slight increase of some antibody isotypes (IgG2a, IgG2b) upon NP-Ficoll immunization (Fig. 1e). These results suggested that Peli1 inhibits B cell antibody production specifically dependent on T cell-mediated signaling.

To further confirm that Peli1-mediated suppression of antibody production is due to its direct function in B cells, we immunized the Rag1-deficient mice that transferred with WT T cells plus WT or KO B cells with NP-KLH. Consistent with the result obtained from global Peli1-KO mice, the production of antigen-specific antibodies, such as IgM, IgG2a, and IgG3, were significantly increased in the recipient mice that transferred with KO B cells, as compared with the mice that transferred with WT B cells ($P < 0.01$, by Student's *t*-tests, Fig. 1f). Thus, these data demonstrated that Peli1 is a negative regulator to restrain the proliferation and antibody production in B cells specifically in response to noncanonical NF-κB stimulation.

**B cell Peli1 protects against the lupus autoimmunity.** In order to examine whether Peli1 may regulate SLE pathogenesis, we induced lupus-like disease in wild-type (WT) and Peli1-knockout (KO) mice by immunization with the CD4$^+$ T cells from BM12 mice. In this model, the donor CD4$^+$ T cells will be recognized by recipient antigen presenting cells and differentiated into T follicular cells (Tfh), which promote the expansion of recipient-derived germinal center (GC) B cells and antibody-producing plasma cells, and the production of autoantibodies[30]. The results showed that WT mice developed clinical symptoms of lupus-like disease that were characterized by the deposition of IgG in the kidney (Fig. 2a), and the immunized Peli1 mutant mice exhibited more severe lupus-like disease with increased IgG deposition in the kidney (Fig. 2a). This phenotype of the KO mice was

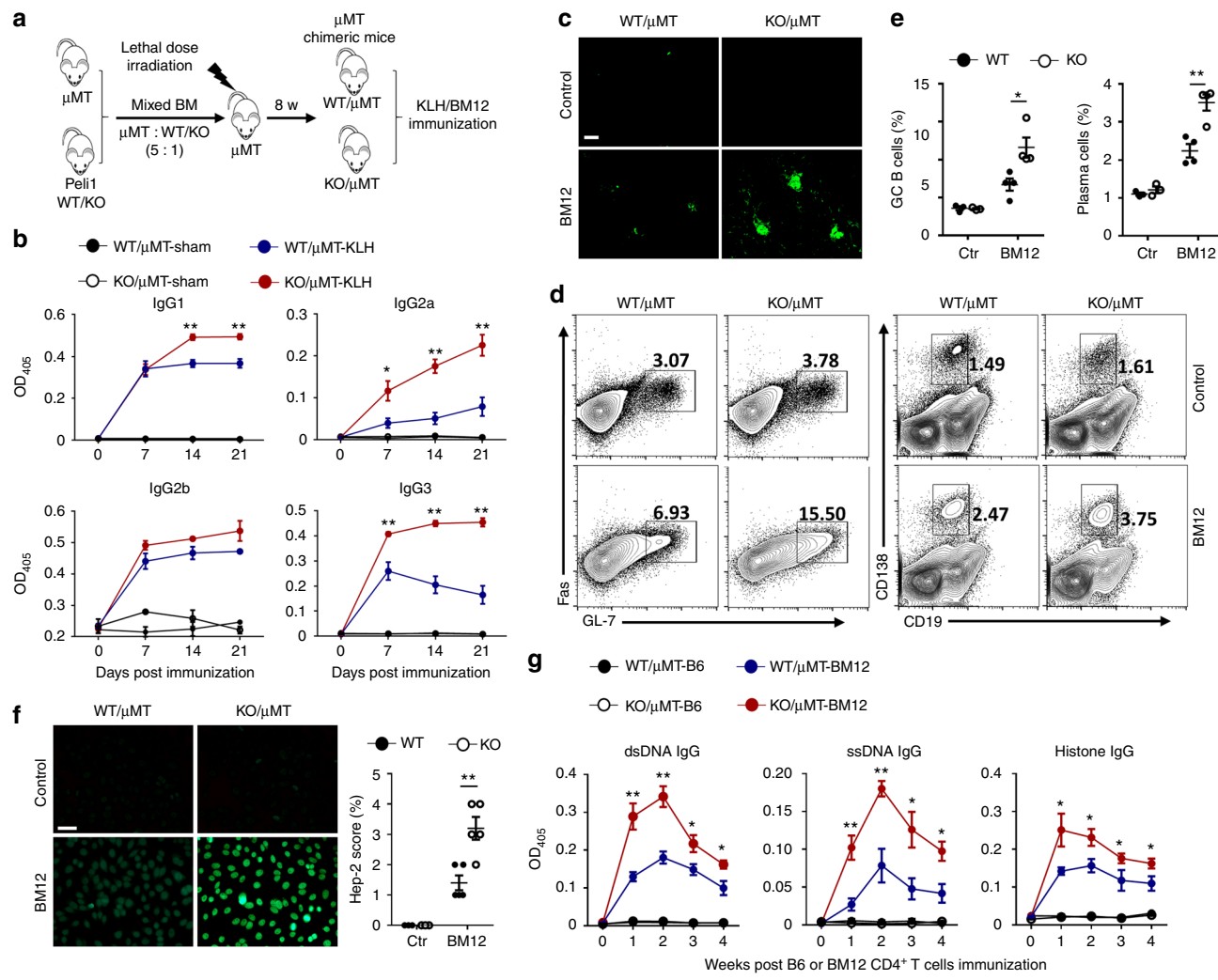

**Fig. 3** Peli1 deficiency specifically in B cells promotes autoimmunity in vivo. **a** Scheme showing how the µMT chimeric mice were constructed for immunization. **b** Enzyme-linked immunosorbent assay (ELISA) of NP-specific IgG1, IgG2a, IgG2b, and IgG3 in the serum of WT/µMT and Peli1-deficient (KO)/µMT chimeric mice that immunized intraperitoneally with vehicle (sham) or NP-KLH (KLH). **c** WT/µMT and KO/µMT chimeric mice were intraperitoneally injected with 7.5 million of CD4 + T cell from C57BL/6 mice (control) or from BM12 mice. Representative immunofluorescent images showing IgG deposits in kidney by staining with Alexa Fluor 488-labeled anti-mouse IgG. Scale bar, 100 µm. **d**, **e** Flow cytometric analysis of the percentages of Fas$^+$GL-7$^+$ germinal center (GC) B cells and CD19$^-$CD138$^+$ plasma cells in immunized WT/µMT and KO/µMT chimeric mice as described in (**c**). Data are presented as the representative FACS plots (**d**) and summary graphs (**e**). **f** Distinct anti-nuclear antibody (ANA) staining patterns of the Hep-2 cell line with serum from WT/µMT and KO/µMT chimeric mice 4 weeks after immunization as described in (**c**). Ctr represents control in the right panel bar graph. Scale bar, 50 µm. **g** Enzyme-linked immunosorbent assay (ELISA) of anti-dsDNA, anti-ssDNA, and anti-histone IgG in serum from immunized WT/µMT and KO/µMT chimeric mice as described in (**c**) at the indicate time point. Data are shown as the mean ± SEM based on three independent experiments. Two-tailed Student's t-tests were performed. *$P < 0.05$ and **$P < 0.01$

associated with increased frequencies of plasma cells and GC B cells (Fig. 2b). Accordingly, the production of serum anti-nuclear antibody, and the serum levels of IgG against dsDNA, ssDNA, and histone were significantly enhanced in KO mice as compared with WT mice ($P < 0.05$ and $P < 0.01$, by Student's t-tests, Fig. 2c, d). Interestingly, we found that BM12 CD4$^+$ T cells immunization significantly induced the activation of noncanonical NF-κB as suggested by increased NIK, the master kinase for this pathway, and the p52 levels in splenic B cells from immunized mice. Moreover, *Peli1* deficiency markedly promoted more NIK and p52 accumulation than that in WT B cells (Fig. 2e), suggested a potential negative role of Peli1 in B cells to regulate noncanonical NF-κB activation and autoimmunity in lupus-like disease.

To confirm that Peli1 in B cell protect against the autoimmunity in lupus-like disease, we constructed the mixed bone marrow (BM) chimeric mice by reconstituting the lethal

dose irradiated B cell-deficient µMT mice with the mixed BMs from µMT mice and Peli1 WT/KO mice (µMT: Peli1 WT/KO = 5: 1) (Fig. 3a). In these chimeric mice, µMT mice BM will provide all the genetic-competent immune cells except for B cells, and Peli1 WT/KO mice BM will provide the Peli1-competent or Peli1-deficient B cells. By using these chimeric mice to mimics the B cell-specific Peli1-KO condition in vivo, we found that NP-KLH immunization induced significantly increased production of serum antigen-specific antibodies in KO/µMT chimeric mice as compared to WT/µMT chimeric mice ($P < 0.01$, by Student's t-tests, Fig. 3b). In addition, KO/µMT chimeric mice developed more severe lupus-like disease than WT/µMT chimeric mice, as characterized by increased IgG deposition in the kidney (Fig. 3c), elevated frequencies of GC B cells and plasma cells (Fig. 3d, e), and enhanced production of serum anti-nuclear antibody (Fig. 3f), and the serum levels of IgG against dsDNA, ssDNA, and histone

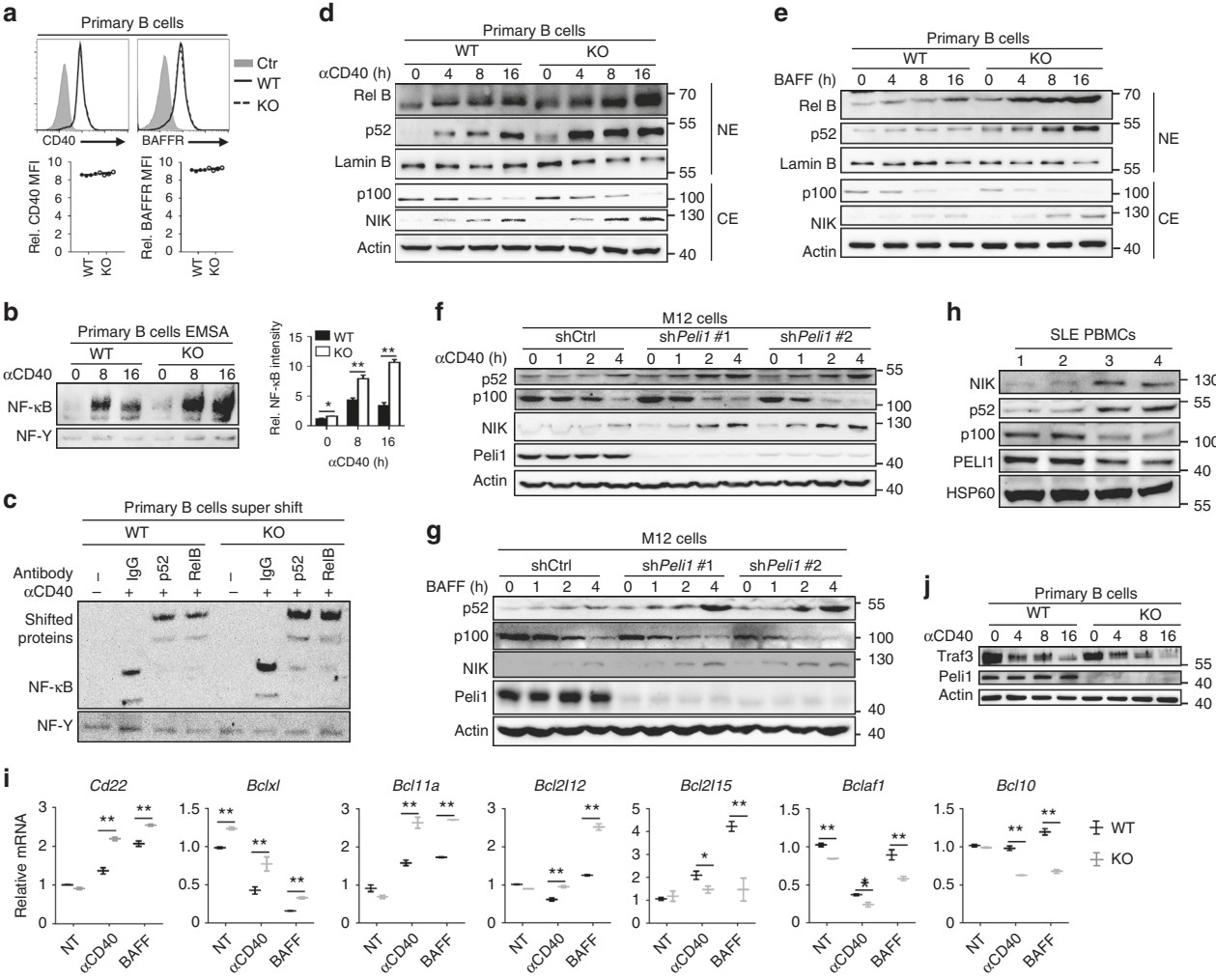

**Fig. 4** Peli1 is a pivotal negative regulator of noncanonical NF-κB pathway. **a** Flow cytometry analysis of the surface expression of CD40 and BAFF receptor (BAFFR) in WT and KO splenic B cells. Data are presented as the representative FACS plots (upper panel) and summary graphs (lower panel). Ctr represents isotype control. MFI = mean fluorescence intensity. **b** Electrophoretic mobility-shift assay (EMSA) of nuclear extracts of WT and KO splenic B cells that left unstimulated or stimulated for 8, 16 h with anti-CD40 (αCD40, 1 μg/ml). Data are presented as the immunoblot panels (left) and the bar graph to quantify the relative NF-κB signals (right). **c** Supershift assay determining the NF-κB complex in the nuclear extracts of WT and KO splenic B cells that left unstimulated or stimulated for 16 h with anti-CD40 (αCD40, 1 μg/ml) by using the control antibody (IgG), anti-p52, or anti-RelB antibody. **d**, **e** Immunoblot analysis of NF-κB proteins, NIK or Actin and lamin B (loading controls) in cytoplasmic extracts (CE) and nuclear extracts (NE) of B cells stimulated with anti-CD40 (αCD40) (**d**) or BAFF (**e**). **f**, **g** Immunoblot analysis of the effect of Peli1 knockdown on NIK and NF-κB p100/p52 protein levels in total lysis of M12 cells that stimulated with anti-CD40 (αCD40) (**d**) or BAFF (**e**). **h** Immunoblot analysis of NIK, p52, p100, Peli1, and HSP60 (loading control) expression in PBMCs of 4 SLE patients. **i** QPCR determining the relative expression of the indicated apoptosis-related genes in WT and Peli1-dificient splenic B cells that left untreated (NT) or stimulated with BAFF or anti-CD40 (αCD40) for 20 h. Data were normalized to a reference gene, Actin. Data are shown as the mean ± SEM based on three independent experiments. Two-tailed Student's t-tests were performed. *P < 0.05 and **P < 0.01. **j** Immunoblot analysis of Traf3 protein levels in WT and KO splenic B cells that stimulated for different time point with anti-CD40 (αCD40)

(Fig. 3g). These results collectively confirmed a B cell-specific role of Peli1 in mediating the autoantibody production and lupus-like autoimmunity, in which Peli1 may mediate the suppression of noncanonical NF-κB activation.

**Peli1 negatively regulates noncanonical NF-κB signaling.** To figure out whether and how Peli1 mediate the inhibition of noncanonical NF-κB activation, we firstly examined the surface expression of CD40 and BAFF receptor (BAFFR), which are the two major receptors responsible for the transduction of non-canonical NF-κB signaling in B cells. We observed equal expression levels of these two receptors on the surface of WT and Peli1-deficient splenic B cells (Fig. 4a). However, electrophoretic

mobility-shift assay (EMSA) results showed that the activation of NF-κB was dramatically promoted in Peli1-deficient splenic B cells that stimulated with anti-CD40 as compared with that in WT B cells (Fig. 4b). In addition, the supershift assay suggested that both anti-RelB and anti-p52 antibody alone shifted nearly the entire NF-κB signals (Fig. 4c), suggesting that anti-CD40-induced translocation of NF-κB complexes were composed primarily of RelB-p52 heterodimers in B cells. Moreover, the heightened NF-κB DNA-binding activity in Peli1-deficient B cells was associated with a marked increase in nuclear levels of p52 and RelB by anti-CD40 and BAFF stimulation (Fig. 4d, e). To exclude the developmental influence on B cell activation, we examined the non-canonical NF-κB activation in Peli1-knockdown M12 B cells, a murine B-lymphoma cells that commonly used to study the

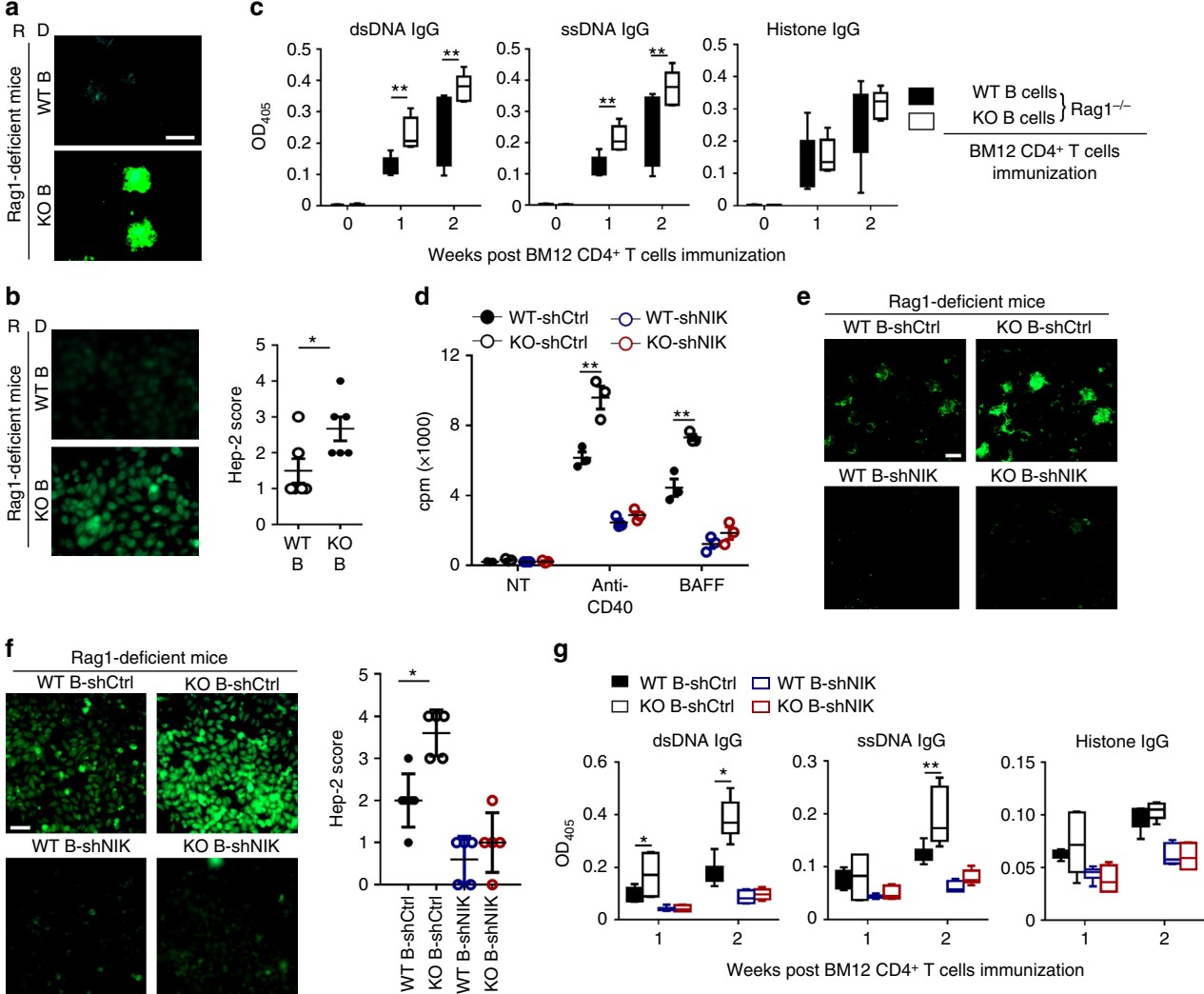

**Fig. 5** The noncanonical NF-κB pathway is required for lupus pathology in Peli1-deficient mice. **a–c** Rag1-dificient mice (recipient, R) were adoptively transferred with WT or KO B cells (donor, D), and then immunized with BM12 CD4$^+$ T cell. The IgG deposits in kidney were determined by staining with Alexa Fluor 488-labeled anti-mouse IgG (**a**, Scale bar, 100 μm), the serum ANA were analyzed by using the Hep-2 cell line (**b**), and the anti-dsDNA, anti-ssDNA, anti-histone IgG in serum were examined by ELISA (**c**). **d** Proliferation of WT and KO splenic B cells that infected with control or NIK shRNA, incubated in vitro for 72 h in the absence (NT) or presence of anti-CD40 or BAFF, then assessed by [$^3$H]thymidine incorporation. **e–g** Rag1-dificient mice were adoptively transferred with WT or KO B cells that infected with control or NIK shRNA, and then immunized with BM12 CD4$^+$ T cell. The kidney IgG deposition were visualized with Alexa Fluor 488-labeled anti-mouse IgG (**e**, Scale bar, 100 μm), the serum ANA were analyzed by using the Hep-2 cell line (**f**, Scale bar, 50 μm.), and the anti-dsDNA, anti-ssDNA, anti-histone IgG in serum were examined by ELISA (**g**). Ctrl, control. Data are shown as the mean ± SEM based on three independent experiments. Two-tailed Student's t-tests were performed. *$P < 0.05$ and **$P < 0.01$

noncanonical NF-κB signaling in B cells. The results showed that both anti-CD40 and BAFF stimulation promoted the activation of noncanonical NF-κB signaling, as suggested by increased NIK accumulation and the p52 levels, along with promoted p100 processing, in *Peli1*-knockdown M12 B cells, as compared with that in control cells (Fig. 4f, g). Interestingly, we also found that the NIK and p52 protein levels were markedly increased, along with decreased p100 protein levels in peripheral blood mononuclear cells (PBMCs) from SLE patients with *Peli1*$^{low}$ (hereafter called SLE-PL) as compared with that from SLE patient with *Peli1*$^{high}$ mRNA expression (hereafter called SLE-PH) (Fig. 4h), suggesting that Peli1 may also negatively regulate noncanonical pathway during human SLE pathogenesis. It is known that noncanonical NF-κB signaling regulates the transcription of apoptosis-related genes[31,32], we also found that *Peli1* deficiency diversely regulated apoptosis-related gene expression in B cells

upon noncanonical NF-κB activation, characterized by increased anti-apoptosis gene expression, whereas decreased pro-apoptosis gene expression in KO cells (Fig. 4i).

In addition to the inhibitory effect of Peli1 in B cells, *Peli1* deficiency also promoted noncanonical NF-κB activation in mouse embryonic fibroblast (MEF) as suggested by increased levels of nuclear p52 and RelB upon the stimulation of anti-LTβR (Supplementary Fig. 2), another well-characterized noncanonical NF-κB inducer[17,33,34]. Accordingly, *Peli1*-deficient mice displayed a marked increase in the size and number of B cell-containing colonic patches (CLPs) (Supplementary Fig. 3a, 3b), and had a significant increase in fecal IgA concentration compared to wild-type mice ($P < 0.01$, by Student's t-tests, Supplementary Fig. 3c). In addition, the intestines of KO mice had elevated expression of two major chemokines, CXCL12 and CXCL13, and the cell adhesion molecule MADCAM1 (Supplementary Fig. 3d), which

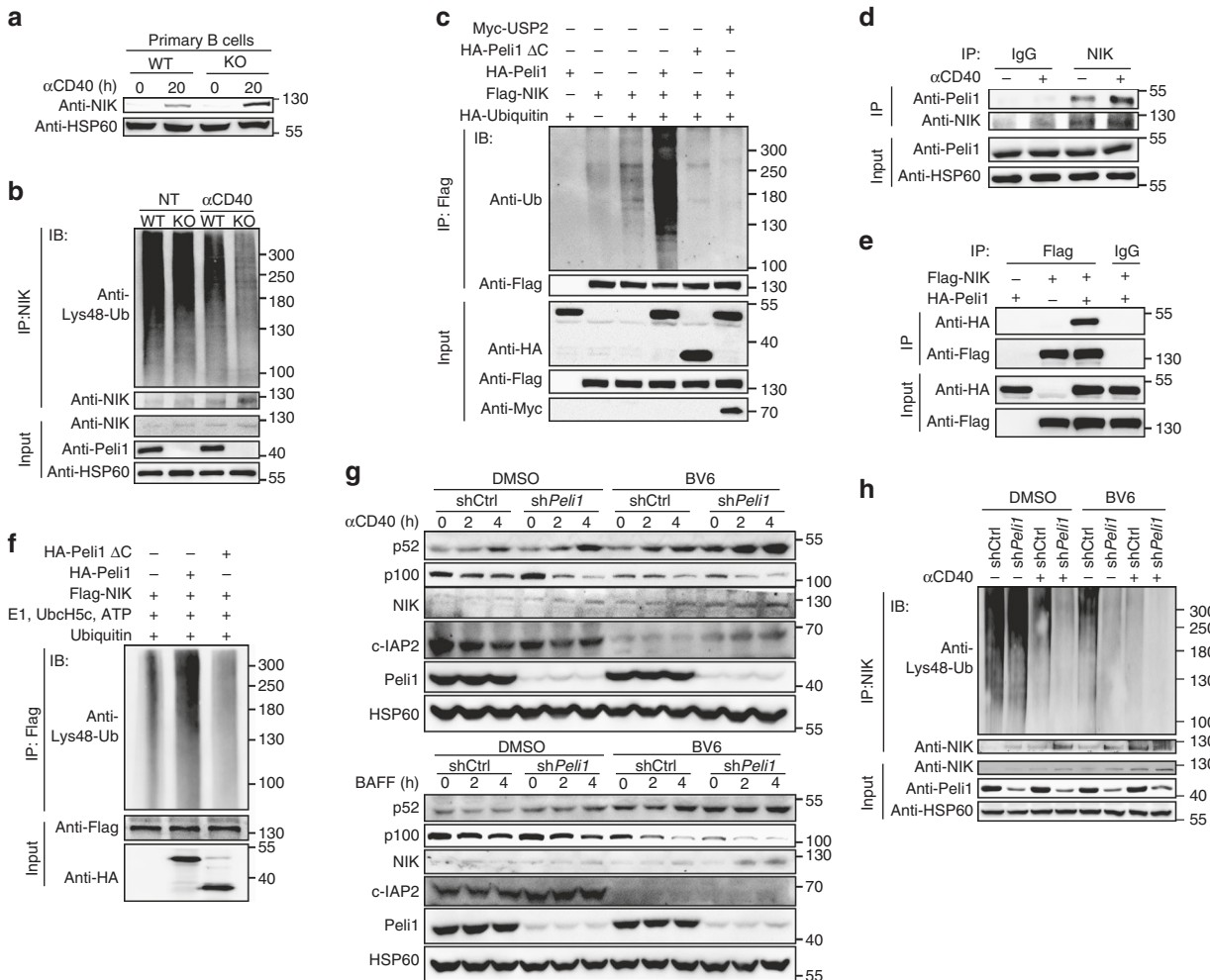

**Fig. 6** Peli1 binds to and mediates Lys48 ubiquitination of NIK. **a** Immunoblot analysis of NIK and HSP60 in whole-cell lysates of WT and *Peli1*-deficient splenic B cells stimulated with anti-CD40 (αCD40). **b** Analysis of Lys48 ubiquitination of NIK in WT and KO B cells left unstimulated or stimulated with anti-CD40 (αCD40) for 4 h in the presence of a proteasome inhibitor MG132. IP, immunoprecipitation; IB, immunoblotting. **c** Ubiquitination of NIK in HEK293 cells transfected with (+) or without (−) indicated expression vectors. **d** Immunoassays on lysates of WT splenic B cells left unstimulated or stimulated with anti-CD40 (αCD40) for 4 h in the presence of MG132, followed by IP with control IgG or anti-NIK and immunoblot analysis of NIK-associated Peli1. **e** IP analysis examining the association of NIK with Peli1 in HEK293 cells transfected with (+) or without (−) indicated expression vectors. **f** Immunoblot analysis of Lys48 ubiquitination of NIK assessed by in vitro ubiquitination assay with a mixture of E1, E2 (UbcH5c), ATP, Flag-NIK, HA-Peli1, or HA-Peli1ΔC after IP with anti-Flag. **g** Immunoblot analysis of p52, p100, NIK, c-IAP2, Peli1, and HSP60 in total lysis of control and *Peli1*-knockdown M12 cells that pretreated with DMSO or smac mimetic BV6 for 4 h, and then stimulated with anti-CD40 (αCD40) (upper panel) or BAFF (lower panel) at the indicated time points. Ctrl, control. **h** Analysis of Lys48 ubiquitination of NIK in control and *Peli1*-knockdown M12 cells that pretreated with DMSO or smac mimetic BV6 for 4 h, then left unstimulated or stimulated with anti-CD40 (αCD40) for 4 h in the presence of MG132. Ctrl, control. Data are shown based on three independent experiments

also confirmed in MEF cells by anti-LTβR stimulation (Supplementary Fig. 3e). These results suggest that Peli1-mediated inhibitory function of noncanonical NF-κB pathway is a universal effect, but not just restrict in B cells.

Activation of noncanonical NF-κB involves signal-induced TRAF3 degradation[23,24,35]. We previously found that Peli1 mediated TLR-induced MAPK activation through the regulation of TRAF3 degradation in microglial cells, which promoted us to examine whether Peli1-mediated inhibition of noncanonical NF-κB pathway is also due to its function on TRAF3. Interestingly, *Peli1* deficiency or knockdown did not affect the TRAF3 degradation status upon noncanonical NF-κB activation either in splenic B cells or in M12 cells (Fig. 4j, Supplementary Fig. 4). Therefore, these data suggested that Peli1 negatively regulates the activation of noncanonical NF-κB signaling without affecting TRAF3 degradation.

**NIK is required for enhanced autoimmunity in *Peli1*-KO mice.** To further confirm the Peli1 function in B cells to regulate SLE pathology, we induced lupus-like disease in Rag1-deficient mice that transferred with WT or KO B cells by immunization with BM12 CD4[+] T cells. In concert with the data from global *Peli1*-deficient mice and μMT chimeric mice, the recipient mice that transferred with KO B cells developed more severe disease, exhibiting increased IgG deposition in the kidney, enhanced ANA production, and increased secretion of autoantibodies against dsDNA and ssDNA (Fig. 5a–c). Thus, these results further established a B cell-intrinsic role of Peli1 to suppress the development of lupus-like disease.

Since the activation of noncanonical NF-κB signaling is dependent on NIK[14,15], we examined whether Peli1-mediated modulation of this pathway is required for NIK. As shown in Fig. 5d, anti-CD40 or BAFF stimulation induced comparable and

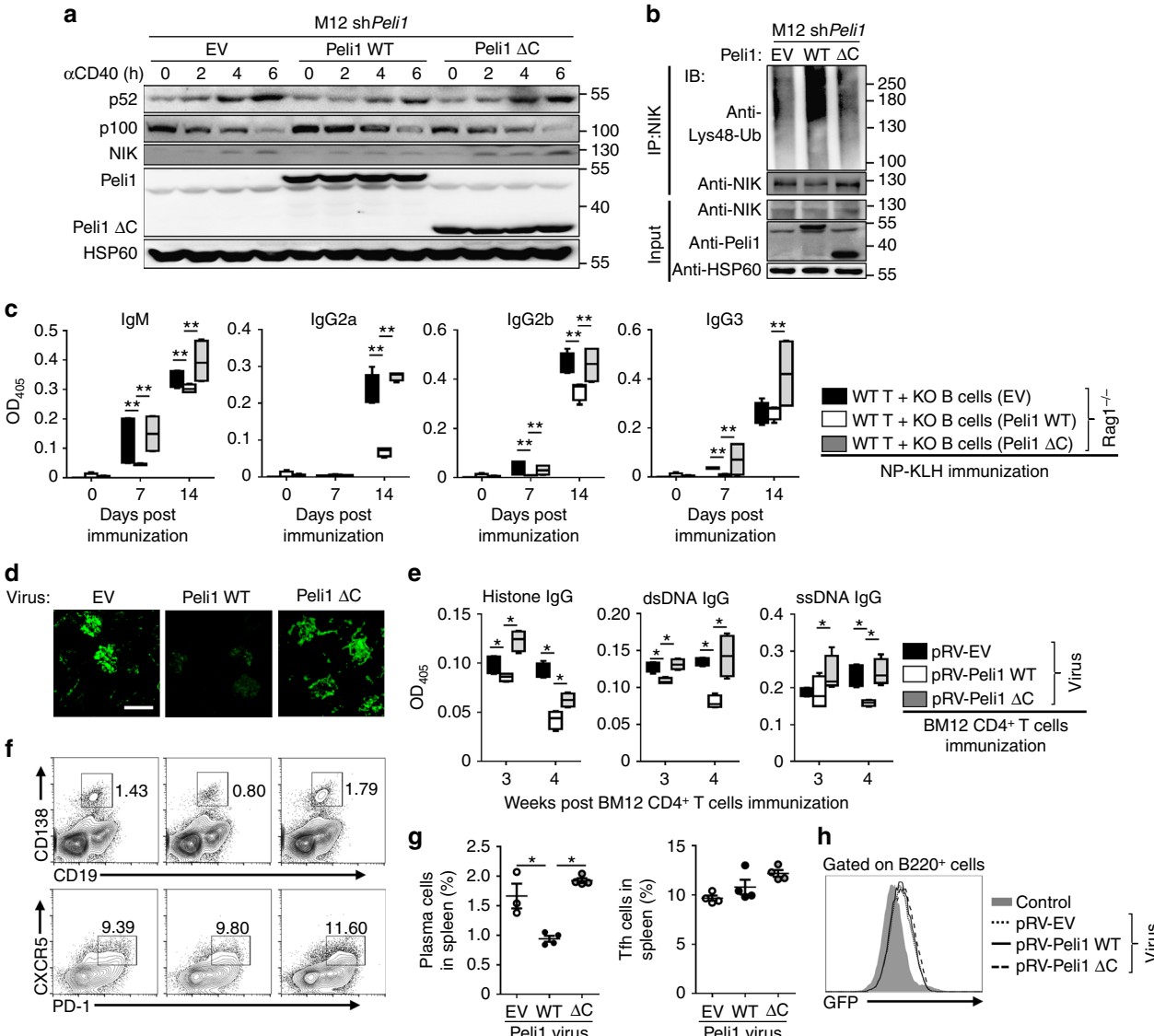

**Fig. 7** Overexpression of Peli1 prevents lupus-like disease. **a** Immunoblot analysis of p52, p100, NIK, Peli1, Peli1ΔC and HSP60 in *Peli1*-knockdown M12 cells that reconstituted with empty vector (EV), WT full-length Peli1 or Peli1ΔC that stimulated with anti-CD40 (αCD40) at the indicated time points. **b** Analysis of Lys48 ubiquitination of NIK in *Peli1*-knockdown M12 cells reconstituted with EV, WT Peli1, or Peli1ΔC that stimulated with anti-CD40 (αCD40) for 4 h in the presence of MG132. **c** LISA of NP-specific IgM, IgG2a, IgG2b and IgG3 in the serum of Rag1$^{-/-}$ mice that transferred with WT T cells plus KO B cells reconstituted with EV, WT Peli1 or Peli1ΔC, and then immunized intraperitoneally with NP-KLH. **d–h** *Peli1*-deficient (KO) mice were immunized with 7.5 million of BM12 CD4$^+$ T cell to induce lupus-like disease, and then injected with pRV-GFP retrovirus encoding Peli1 or Peli1ΔC 12 h post-immunization. The IgG deposits in kidney were examined by staining with Alexa Fluor 488-labeled anti-mouse IgG (**d**). Scale bar, 100 μm. The anti-dsDNA, anti-ssDNA, anti-histone IgG in serum were determined by ELISA (**e**). The percentages of CD19$^-$CD138$^+$ plasma cells, PD-1$^+$CXCR5$^+$ Tfh cells (**f**, **g**) and infected B220$^+$ B cells (**h**) in spleens were assessed by flow cytometry. Data are shown as the mean ± SEM based on three independent experiments. Two-tailed Student's *t*-tests were performed. *$P < 0.05$ and **$P < 0.01$

suppressed proliferation index between WT and KO B cells that were knocked down of NIK, as compared with control cells. Furthermore, NIK knockdown in B cells alleviated the lupus-like disease, and abolished the phenotype differences in Rag1-deficient immunized mice that transferred with WT or KO B cells, characterized by suppressed comparable kidney IgG deposition, serum production of ANA and autoantibodies against dsDNA and ssDNA (Fig. 5e–g). Taken together, NIK is required for the promoted activation of noncanonical NF-κB signaling, and the enhanced lupus-like autoimmunity in *Peli1*-KO mice.

**Peli1 mediates Lys48-ubiquitination of NIK.** Noncanonical NF-κB activation requires the accumulation of its master kinase

NIK[23,24,36], which is regulated by its Lys48-linked ubiquitination and degradation. We found that *Peli1* deficiency or knockdown significantly promoted NIK accumulation and attenuated Lys48 ubiquitination of NIK in primary splenic B cell (Fig. 6a, b) and M12 B cells (Supplementary Fig. 5a). In addition, overexpression of full-length Peli1, but not its RING-deletion mutant (Peli1ΔC, loss of E3 ligase function) markedly enhanced NIK ubiquitination, and overexpression of a non-specific deubiquitinase USP2 almost hydrolyzed all the ubiquitin chains of NIK that induced by full-length Peli1 (Fig. 6c, Supplementary Fig. 5b, 5c), suggesting Peli1 indeed mediate NIK ubiquitination but not the other modifications. Because the ubiquitination process requires the association of E3 ligase with its substrate, we examined whether

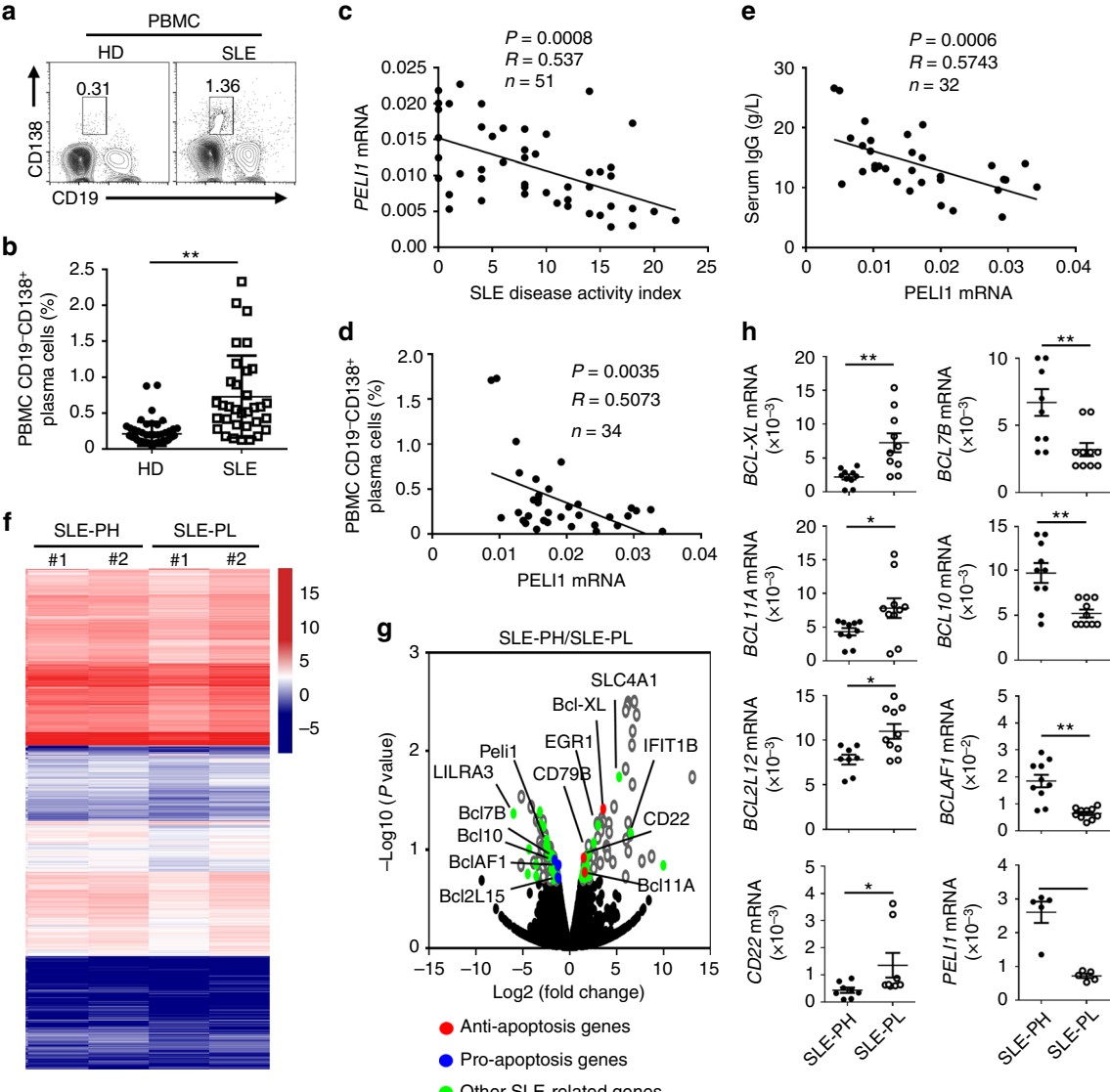

**Fig. 8** PELI1 expression is negatively correlated with SLE disease activity. **a, b** Flow cytometric analysis of the frequencies of CD19⁻CD138⁺ plasma cells in peripheral blood mononuclear cells (PBMCs) from health donors (HD) and SLE patients. Data are presented as a representative plot (**a**) and summary graph (**b**). **c–e** Correlation analysis of the *PELI1* mRNA expression in PBMCs with the disease activity (**c**), plasma cells frequencies (**d**), and serum IgG levels (**e**) in SLE patients. **f** Heat map showing differentially expressed genes in PBMCs that isolated from SLE patients with *Peli1*^high or *Peli1*^low mRNA expression (SLE-PH or SLE-PL). **g** Volcano graphs showing the genes differentially expressed in PBMCs between SLE-PH and SLE-PL. **h** QPCR determining the relative expression of the indicated apoptosis-related and *Peli1* genes in PBMCs of SLE-PH and SLE-PL patients. Each dot represents a value of the gene epression results from SLE-PH or SLE-PL patients. Data were normalized to a reference gene, *ACTIN*. Data are shown as the mean ± SEM based on three independent experiments. Two-tailed Student's *t*-tests were performed. *$P < 0.05$ and **$P < 0.01$

Peli1 directly target NIK for ubiquitination. Expectedly, Peli1 and NIK form a complex in resting cells, with an enhanced association between these two proteins upon anti-CD40 stimulation (Fig. 6d, Supplementary Fig. 5d). The Peli1–NIK interaction was further confirmed using a transient transfection experiment in which the overexpressed Peli1 strongly associated with NIK (Fig. 6e). Moreover, in vitro ubiquitination assay indicated that WT full-length Peli1, but not its RING-deletion mutant mediated Lys48 ubiquitination of NIK in the presence of E1, E2 (UbcH5c), and ATP (Fig. 6f). These results suggested that Peli1 directly targets NIK and mediates Lys48 ubiquitination and degradation of NIK.

Published results have demonstrated TRAF2-cIAPs E3 complex is the only previously known regulatory system for Lys48 ubiquitination of NIK[23,24], we asked whether Peli1-mediated NIK

Lys48 ubiquitination requires cIAPs. Consistent with the reported data, cIAPs inhibition by its specific inhibitor smac mimetic BV6 promoted NIK accumulation, p100 processing as well as increased protein levels of p52. Interestingly, *Peli1* knockdown further enhanced smac mimetic-mediated incensement of NIK and p52 level either by anti-CD40 or BAFF stimulation (Fig. 6g). Moreover, *Peli1* knockdown could further attenuated Lys48 ubiquitination of NIK even without cIAPs (Fig. 6h). Collectively, these data suggested Peli1 serves as a direct E3 ligase of NIK, and mediates Lys48 ubiquitination and degradation of NIK independent of cIAPs.

**Overexpression of Peli1 prevents lupus-like disease**. Since the aforementioned results established Peli1 as a negative regulator of NIK accumulation, we reasoned that overexpression of Peli1 may

has adverse effect for noncanonical NF-κB activation. Indeed, overexpression of WT Peli1 significantly inhibited p100 processing to p52, along with promoted Lys48 ubiquitination of NIK. In contrast, overexpression of Peli1ΔC has no effect on either p100 processing or NIK ubiquitination (Fig. 7a, b), suggesting the E3 ligase activity of Peli1 is critical for its suppressive function on noncanonical NF-κB pathway. In addition, the primary splenic KO B cells that reconstituted with WT Peli1 produced much less antibody like IgM, IgG2a, IgG2b, and IgG3 in response to NP-KLH immunization, as compared with that reconstituted with either empty vector (EV) or Peli1ΔC (Fig. 7c). These data promoted us to assume that Peli1 may has therapeutic effect on lupus-like disease, we then injected the retrovirus encoding WT Peli1 or Peli1ΔC into BM12 CD4$^+$ T cells-immunized mice. As expected, retrovirus encoding WT Peli1 significantly attenuated the severity of lupus-like disease, as suggested by inhibited IgG deposition in the kidney, suppressed production of serum autoantibodies against dsDNA, ssDNA, and histone, and decreased frequencies of plasma cells along with unaltered percentages of T follicular helper (Tfh) cells (Fig. 7d–g). Together these findings suggested a potential function of Peli1 in the treatment of lupus-like disease through the inhibition of noncanonical NF-κB activation.

**PELI1 levels negatively correlate with human SLE.** Published study demonstrated that *PELI1* polymorphisms is genetically associated with SLE susceptibility in Chinese population[37], we then collected PBMCs from healthy donors (HD) and SLE patients to investigate the potential role of PELI1 in regulating B cell-mediated immune response and SLE pathogenesis. Under healthy conditions, there are very few antibody-producing plasma cells detected in HD PBMCs. In contrast, the frequencies of plasma cells in PBMCs are significantly increased under SLE pathological condition ($P < 0.01$, by Student's $t$-tests, Fig. 8a, b). Interestingly, *PELI1* mRNA levels in SLE patients are negatively correlated with the disease activity (Fig. 8c), and individuals with lower levels of PBMCs *PELI1* mRNA are also associated with higher percentage of antibody-producing CD19$^-$CD138$^+$ plasma cells and serum IgG levels among SLE patients (Fig. 8d, e).

To dissect the mechanistic role of PELI1 in regulating human SLE pathogenesis, we conducted gene expression profile analysis by using PBMCs isolated from SLE-PH and SLE-PL (Fig. 8f). The volcano plot showed that a large number of genes that reported to regulate SLE pathology, especially apoptosis-related genes, such as CD22, BCL-XL, and BCL10, etc., were detected among the differentially expressed genes in PBMCs between SLE-PH and SLE-PL (Fig. 8g). In addition, real-time quantitative PCR confirmed the expression alteration of these apoptosis-related genes in PBMCs between SLE-PH and SLE-PL (Fig. 8h), which is consistent with the gene expression profiles in *Peli1*-deficient B cells that stimulated by noncanonical NF-κB inducer. Together these results suggested PELI1 is negatively correlated with human SLE pathogenesis, which maybe attributed to the alteration of noncanonical NF-κB activation.

**Discussion**
Uncontrolled production of pathogenic autoantibodies is the hallmark of SLE, which centralizes B cells as a key player in the regulation of this autoimmune disease[5–7,38]. The noncanonical NF-κB pathway critically regulates B cell activation and antibody production. Thus, therapeutic strategies based on the inhibition of noncanonical NF-κB activation have been applied for the treatment of SLE. For example, Belimumab, an FDA approved drug for SLE treatment, is a monoclonal antibody that antagonizes the noncanonical NF-κB inducer BAFF[8–12]. However,

Belimumab exhibits low efficacy for SLE treatment, we reasoned that it cannot totally blocks noncanonical NF-κB activation, since non-neutralized CD40L and TACI are also two major inducers for noncanonical pathway. So, targeting NIK, the master kinase to activate noncanonical NF-κB pathway, would be an ideal approach for SLE treatment. In the present study, we identified the E3 ubiquitin ligase Peli1 directly target NIK and induce its degradation. As a consequence, *Peli1* deficiency specifically promoted mouse splenic B cell proliferation and antibody production. In addition, *Peli1*-deficient mice developed more severe lupus-like disease along with standard clinical symptoms of SLE. In human SLE patients, *PELI1* low expression in PBMCs was associated with more severe disease activity and increased production of autoantibodies. Thus, here we demonstrated a protective role of PELI1 in SLE pathology through inhibiting noncanonical NF-κB activation and B cell antibody production.

Noncanonical NF-κB pathway that activated by BAFF or anti-CD40 controls B cell survival and activation. NIK is the master kinase in this pathway that responsible to transduce downstream signals from BAFFR and CD40, resulting in p100 processing and p52/RelB translocation into nucleus[17,32]. Conditional deletion of NIK in mice resulted in a reduction in mature class-switched B cells, and NIK inactivation caused a defect in antibody production[18,19]. By using the *Peli1*-deficient mice, here we found that Peli1 is a negative regulator of noncanonical NF-κB pathway in B cells through modulation of NIK levels. Accordingly, *Peli1*-deficient mice produced more antibodies, especially promoted the IgG2a class-switch, by NP-KLH immunization. We speculated that Peli1 may regulate T cell-dependent STAT1 or T-bet activation, which are both critical for IgG2a class-switch[39,40], to cooperate with noncanonical NF-κB signaling in controlling IgG2a induction in B cells. It is reported that the accumulation of NIK is required for TRAF3 degradation[41]. However, our data revealed that Peli1 didn't affect TRAF3 degradation upon stimulation by noncanonical inducers, suggesting a direct regulation of NIK protein level by Peli1. Indeed, we confirmed that Peli1 directly bound to NIK, and mediated the Lys48-linked ubiquitination and degradation of NIK.

It is reported that TRAF2-cIAPs is the only previously known E3 ligase complex that mediate the Lys48 ubiquitination of NIK[23,24,42]. In the present study, we identified Peli1 as another new E3 ligase directly mediated the Lys48 ubiquitination and degradation of NIK. *Peli1* deficiency significantly inhibited signal-induced Lys48 ubiquitination of NIK and promoted NIK accumulation, along with enhanced noncanonical NF-κB activation. In addition, Peli1 expression in HEK293T cells led to potent induction of NIK ubiquitination. As the ubiquitin ligase function of Peli1 required its C-terminal RING domain, the Peli1ΔC mutant largely lost its ability to induce NIK ubiquitination. In addition, it is reported that Peli1 may function as an adaptor that independent of its E3 ligase activity[43], so future studies are needed to investigate more possible unknown function of Peli1. We previously demonstrated that Peli1 is required for TLR-induced cIAPs ubiquitination and activation in microglia[29], so it is reasonable to assume that Peli1-mediated NIK ubiquitination maybe due to the activation of cIAP by Peli1. Interestingly, in vitro ubiquitination assay suggested a direct function of Peli1 to NIK ubiquitination in a cell-free system. Moreover, Peli1 knockdown could further attenuate the NIK ubiquitination even in the absence of cIAPs, suggesting a specific function of Peli1 in different cell types and in response to different stimuli.

We also observed that the Lys48-linked ubiquitination of NIK was slightly reduced in the resting Peli1-deficient B cells as compared with WT cells, whereas anti-CD40 stimulation dramatically impaired the Lys48-linked ubiquitination of NIK in Peli1-deficient B cells. These results suggested that Peli1 maybe

not fully activated to function as an E3 ligase for NIK ubiquitination until upon the stimulation of noncanonical NF-κB inducers. Previous study has identified TBK1 as a critical protein kinase for Peli1 phosphorylation and activation[44], and we have also suggested that TBK1 is activated in B cells upon the stimulation of noncanonical inducers[36]. Thus, it is reasonable to speculate that TBK1 is the key protein kinase that responsible for the phosphorylation and activation of Peli1, which is then activated to mediate the Lys48-linked ubiquitination and degradation of NIK in noncanonical NF-κB signaling.

SLE pathogenesis involves the activation of innate immune cells, which produce proinflammatory cytokines to mediate the autoimmune inflammation[42,45]. We and others previously reported that Peli1 mediates TLR-induced canonical NF-κB activation and proinflammatory gene expression in innate immune cells[27,29]. Indeed, Peli1 deficiency impairs LPS-induce B cell proliferation. Additionally, Peli1 is dispensable for BCR-mediated B cell activation, suggesting increased production of autoantibodies and B cell activation during SLE pathogenesis in Peli1-deficient mice is attributing to the promoted activation of noncanonical NF-κB pathway but not from TLR or BCR signaling. Published study suggested microRNA-155 targets Peli1 to control the generation and function of Tfh cells[46], another important player during SLE pathogenesis. Considering our previous data that Peli1 negatively regulates T cell proliferation and activation by targeting c-Rel ubiquitination[28], it is reasonable to assume that Peli1 may also function in T cells to mediate the autoimmune inflammation during SLE pathogenesis, and explains the observed phenotype that the difference of lupus-like disease between WT and Peli1 global KO mice is more obvious than that in Rag1-deficient mice transferred with WT T plus WT or KO B cells.

Our data also provided clinical evidences that PELI1 expression in PBMCs is negatively associated with SLE disease activity and the production of autoantibodies, suggested that PELI1 could be developed as a biomarker for SLE diagnosis. In addition, lower expression of PELI1 suggested more severe disease activity, implying worse outcome of SLE prognosis. Moreover, Peli1-deficient mice developed more severe lupus-like disease, whereas overexpression of Peli1 significantly inhibited noncanonical NF-κB activation, which in turn inhibited antibody production and suppressed lupus-like disease. Thus, enhancing PELI1 expression could be a potential therapeutic strategy for SLE treatment.

In summary, our work establishes Peli1 as a mediator of noncanonical NF-κB activation in B cells and SLE pathogenesis. Based on our data, we propose a model in which Peli1 function as an E3 ligase to directly mediate Lys48 ubiquitination and degradation of NIK, thereby restricting the activation of noncanonical pathway and autoimmune inflammation in lupus-like disease (Supplementary Fig. 6). Since accumulating evidences suggest the critical function of noncanonical NF-κB signaling in various diseases like SLE, therapeutic strategies by targeting Peli1 maybe beneficial for the treatment of related diseases.

## Methods

**Patients samples.** Fifty-three adult female health donors (HD) and 51 adult female SLE patients who met the American College of Rheumatology revised criteria for SLE were enrolled. All HD or SLE patients' samples were used after written informed consent was obtained. PBMCs isolated form HD or SLE patients' blood samples were subjected to flow cytometry, immunoblot and real-time quantitative PCR analysis. The clinical disease activity was assessed according to the SLE Disease Activity Index (SLEDAI).

**Mice.** Peli1-defcient mice (on a C57BL/6 background) were provided by Dr. S. Sun[27] (The University of Texas M.D. Anderson Cancer Center, Houston, TX). Peli1[+/−] mice were bred to generate Peli1[−/−] (KO) and Peli1[+/+] (WT) mice, which were used in the experiments. BM12 transgenic mice and μMT mice were

provided by Dr. N. Shen[47] (Shanghai Institutes for Biological Sciences, Chinese Academy of Sciences). Rag1[−/−] mice (NM-KO-00069) were purchased from Shanghai Model Organisms Center. All mice were maintained in a specific pathogen–free facility, and were sacrificed by cervical dislocation for animal experiments, and all animal procedures were approved by the institutional Biomedical Research Ethics Committee, Shanghai Institutes for Biological Sciences, Chinese Academy of Sciences.

**Plasmids and reagents.** The plasmid pcDNA-HA-Peli1/Peli1ΔC, pcDNA-HA-Traf3, pcDNA-HA-cIAP2, pcDNA-Flag-NIK, PRV-Peli1/Peli1ΔC, pLKO.1-shPeli1, pGIPZ-shNIK were provided by Dr. S. Sun[27–29,36]. The Myc-USP2 plasmid was provided by Dr. B. Li[48]. The anti-Peli1 (F-7, SC-271065), anti-Peli1-HRP (F-7, SC-271065 HRP), anti-p100/p52 (C-5, SC-7386), anti-RelB (C-19, SC-226), anti-Traf2 (C-20, SC-876), anti-Traf3 (H-122, SC-1828), anti-lamin B (C-20, SC-6216), and anti-Ubi (P4D1, SC-8017) were purchased from Santa Cruz Biotechnology. Anti-actin (AC-74, A2228) and anti-Flag (M2, F3165) were purchased from Sigma. Anti-cIAP2 was purchased from R&D System. Anti-Lys48 ubiquitin (051307) was purchased form Millipore. Anti-mouse CD40 (553788) was from BD Biosciences. BAFF (phc1674) was from Biosource. NP-Ficoll (F-1420), NP-KLH (N-5060), and NP-LPS (N-5065) were purchased from Biosearch Technologies. FITC conjugated anti-CD21 (8D9, 11-0211-82), PE conjugated anti-CD23 (B3B4, 12-0232-82), PB conjugated anti-IgM (eB121-15F9, 48-5890-82), APC conjugated anti-IgD (11-26c, 17-5993-82), PE-cy7 conjugated anti-CD19 (1D3, 25-0193-82), PE conjugated anti-CD138 (Syndecan-1, 142504), PB conjugated anti-GL-7 (48-5902-82), APC conjugated anti-Fas (15A7, 17-0951-82), PE conjugated anti-human CD138 (DL-101, 12-1389−42), FITC conjugated anti-human CD19 (SJ25-C1, 11-0199−42), and anti-mouse LTβ (3C8, 165671-82 for MEF stimulation) were purchased from eBioscience. Mouse CD4 (L3T4, 130-049−201) and CD45R (B220, 130-049-501) MicroBeads were purchased from Miltenyi Biotec. Smac mimetic BV6 (HY-16701) was from MCE. MG132 (C2211) and LPS (L3129) were from Sigma. The anti-IgM (115-006-075 for B cell stimulation) were from Jackson ImmunoResearch.

**Cell culture.** The human embryonic kidney 293 T cells[29] was cultured with DMEM containing 10% FBS. Cells were seeded in six-well plates and were transfected by the LipoFiter method. The mouse M12 cells (M12.4.1)[35] were cultured with RIPM medium supplemented with 10% FBS. To prepare Peli1[+/+] and Peli1[−/−] primary MEFs, Peli1[+/−] mice were bred to generate Peli1[+/+] and Peli1[−/−] embryos. At day 13.5 embryos from the same pregnant female were used to prepare MEFs. The cells were cultured in DMEM supplemented with 10% FBS. Primary B cells were isolated from the splenocyte samples by anti-B220-conjugated magnetic beads (Miltenyl Biotec) and cultured in RIPM medium supplemented with 10% FBS. For B cell proliferation assay, $10^5$ B cells were seeded in 96-well plates with 3 replicates and then stimulated for a total 72 h by various agents. After stimulation, B cells were labeled with [³H]thymidine 8 h before examination.

**Mouse immunization and antibody detection.** Age (6–7 week) and sex-matched WT and Peli1-KO mice were immunized with 4-hydroxy-3-nitrophenylacetyl (NP)-keyhole lympet hemocyanin (KLH), NP-Ficoll or NP-LPS (100 μg for each antigen). After 0, 7, 14, or 21 days later, the sera were collected and analyzed by enzyme-linked immunosorbent assay (ELISA). To generate a lupus-like disease mouse model, 7.5 million of purified CD4[+] BM12 T cells were intraperitoneally injected into recipient mice. Sera were collected at the indicated time points to examine anti-dsDNA, anti-ssDNA, and anti-histone antibodies by ELISA. Hep-2 cells were used to detect anti-nuclear antibodies. Formalin-fixed frozen mouse kidney sections were stained with Alexa Fluor 488-conjugated goat anti-mouse IgG (Invitrogen). Antibody staining was detected using a fluorescent microscope (ZEISS).

**BM chimera.** We adoptively transferred lethally irradiated (¹³⁷Cs, γ-ray, 950 rad) μMT mice (6–8 weeks old) with the mixed BMs from μMT mice and Peli1 WT/KO mice (μMT: Peli1 WT/KO = 5: 1). Under these conditions, μMT mice BM will provide all the genetic-competent immune cells except for B cells, and Peli1 WT/ KO mice BM will provide the Peli1-competent or Peli1-deficient B cells. After 8 weeks, the chimeric mice were intraperitoneally immunized with 100 μg NP-KLH or CD4[+] BM12 T cells as described above.

**Adoptive transfer.** Purified WT or Peli1-KO B cells mixed with WT naïve T cell were transferred into Rag1-deficient mice intravenously. Twenty-four hours later, the bmice were intraperitoneally immunized with 100 μg NP-KLH or CD4[+] BM12 T cells as described above.

**Peli1 or NIK knockdown and Peli1 reconstitution.** HEK293 cells were transfected with pLKO.1-shCtrl, pLKO.1-shPeli1, pGIPZ-shCtrl, or pGIPZ-shNIK along with packaging vectors pMD2 and psPAX2. The lentiviral supernatants were collected 48 h later and used for M12 cell or primary B cell infection and subsequent selection (puromycin for Peli1 knockdown, GFP for NIK knockdown) as described previously[36]. For Peli1 reconstitution, pRV-GFP retrovirus encoding Peli1 or Peli1ΔC were used to transduced Peli1-deficient primary B cells or M12 Peli1-

knockdown cells. The infected cells were then enriched by flow cytometric sorting on the basis of GFP expression.

**Immunoblot and ubiquitination assay.** Purified B cells and M12 cells were left unstimulated or stimulated for appropriate time by anti-CD40 or BAFF. Total or subcellular extracts were prepared for immunoprecipitation or immunoblot analysis with specific antibodies.

For in vivo ubiquitination assays, the B cells were pretreated with MG132 for 1 h and left unstimulated or stimulated with anti-CD40 for 4 h, and then were lysed with cell lysis buffer containing protease inhibitor and N-ethylmaleimide. The cell extracts were boiled for 5 min in the presence of 1% SDS to dissociate the NIK-interacting proteins, and then were diluted with lysis buffer till the concentration of SDS was 0.1% before immunoprecipitation. NIK was then immunoprecipitated from the cell extracts and the immunoprecipitates were immunoblotted with anti-ubiquitin or anti-Lys48 ubiquitin antibody.

For in vitro ubiquitination assay, Flag-NIK, HA-Peli1, and HA-Peli1ΔC were translated in vitro with the TNT® Quick Coupled Transcription/Translation Systems (Promega). Ubiquitination reactions were processed with Ubiquitinylation kit (ENZO) according to the manufacturer's instructions. After 4 h of incubation at 37 °C, the reactions were terminated by boiling for 5 min in SDS sample buffer. The samples were subjected to SDS-PAGE, and followed by immunoblot analysis to examine Lys48 ubiquitination of NIK.

**EMSA and supershift assay.** Nuclear extracts were prepared and subjected to EMSA analysis with a biotin-labeled control probe bound by NF-Y (5′-AAGA-GATTAACCAATCACGTACGGTCT-3′) or a κB oligonucleotide probe (5′-CAACGGCAGGGGAATTCCCCTCTCCTT-3′), by using the LightShift® Chemiluminescent EMSA Kit (Thermo). For supershift analysis, 2 μl of antibody against p52 or RelB was added to the nuclear extract 15 min before the labeled probe was added.

**Flow cytometry.** Single-cell suspensions were stained with antibodies against different cell surface markers, followed by incubation for 25 min on ice, and then the cells were resuspended in PBS with 2% FBS for flow cytometry analysis. For the gating strategy, FSC/SSC is initial applied, and then used the antibodies with specific fluorochrome to make the subsequent gates. All the samples in the same experiments and comparisons were gated under the same parameters.

**RNA-sequencing analysis.** PBMCs isolated from female HD or SLE patients (with high or low *Peli1* expression) were applied for total RNA extraction with TRIzol (Invitrogen) and subjected to RNA-sequencing analysis. RNA sequencing was performed by BGI Tech Solutions. The raw reads were mapped to the mm10 reference genome (build mm10), using Bowite. Gene expression levels were quantified by the RSEM software package. The significantly affected genes were acquired by setting a fold change ≥2 and a false discovery rate threshold of 0.001. Differentially expressed genes were analyzed by the IPA and DAVID bioinformatics platform.

**Quantification and statistical analysis.** Except where otherwise indicated, all the presented data are representative results of at least three independent repeats. Data are presented as mean ± SEM, and the $P$-values were determined by two-tailed Student's $t$-tests. A $P$-values less than 0.05 is considered statistically significant.

**Data availability.** The RNA-Sequencing data have been deposited into the Gene Expression Omnibus (accession code GSE101437). Source data files for Fig. 8 are available online. All other data supporting the findings of this study are available from the corresponding author on reasonable request.

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

## Acknowledgements

This research was supported by the grants from the National Natural Science Foundation of China (81571545, 81770567), the Jiangsu Provincial Key and Development Program (BE2016722), the Thousand Young Talents Plan of China, the US National Institutes of Health (AI104519), CAS Key Laboratory of Stem Cell Biology, and Collaborative Innovation Center of Systems Biomedicine.

## Author contributions

J.L. designed and performed the experiments, prepared the figures, and wrote part of the manuscript; X.H. provided the clinical samples and contributed to the experiments; S.H., Y.W., M.L., J.X., Xingli.Z., T.Y., S.G., D.D., X.L., Q.L., C.M., Y.Z., M.H., Xiaodong.Z. and J.J. contributed to part of the experiments; B.L. provided the USP2 plasmid; N.S. provided the μMT and BM12 transgenic mice; X.C. and S-C.S. provided the Peli1-KO mice and expression vectors; Y.X. designed and supervised the work, prepared the figures, and wrote the manuscript.

## Additional information

**Competing Interests:** The authors declare no competing interests.

