## [Peer Review File(PDF 411 kb) · Nature Communications]

Reviewers' comments:

Reviewer #1 (NFkB, TF regulation)(Remarks to the Author):

In this study, Liu et al identify the protein kinase NIK as a substrate of the E3 ubiquitin ligase Pellino-1. They show that Pellino-1 binds to and mediates proteasomal K48-ubiquitination-linked degradation of NIK. By controlling NIK protein levels Pellino-1 plays a critical role in preventing activation of NIK and of the non-canonical NF- κ B pathway in anti-CD40- or BAFF-mediated B cell activation. A second part of the study shows that loss of Pellino-1 enhances B-cell dependent autoantibody production and the development of systemic lupus erythematoses symptoms in a mouse model. This effect is supported by a negative correlation between Pellino-1 mRNA levels in plasma cells and SLE disease activity in samples obtained from human patients. These results define a novel interesting function of Pellino-1 and relate it to an autoimmune disease for which no clear mechanism has been firmly established. The results are carefully presented and match the author's conclusions. The evidence for the direct role of Pellino-1 in regulation of NIK is strong and supported by several independent approaches, including the usage of catalytically inactive Pellino-1 mutants, overexpression experiments and in vitro ubiquitination assays. The role of the Pellino-1-NIK complex in regulation of RelB/p52-dependent genes as suggested in the summary figure S7 lacks sufficient experimental support, e.g. ChIP-PCR assays showing that RelB/p52 dimers bind to anti-CD40- or BAFF-induced target genes in a NIK- or Pellino-1-dependent manner.

Specific points:

Fig. 3A: There is already a weak NF- κ B/DNA complex visible in nuclear extracts of unstimulated Pellino-1 ko cells. Quantification from replicates of the bandshift experiment should be shown to judge the enhancement of basal versus inducible DNA binding of NF- κ B subunits in cells deficient for Pellino-1. Supershifts using anti RelB or p52 antibodies should be included to show activation of the heterodimer as proposed in Fig. S7.

Fig.5: The amounts of NIK in input and pulldown samples need to be shown in the blot panels of Fig.5B and 5H. Negative controls (e.g. IgG IPs) need to be included in the IP experiments shown in Fig. 5D, E.

Fig. 6B. Likewise, the NIK blot of the input and IP samples needs to be shown to verify that comparable amounts of NIK were immunoprecipitated.

Minor points:

Molecular mass markers should be shown for all immunoblot panels.

The manuscript requires proofreading as it contains several typos and grammatical errors, e.g. in the legend of Fig. 3 (...in total lysis (lysates) of M12 cells that (were) stimulated with.....)

Reviewer #2 (NFkB, Pellino)(Remarks to the Author):

The manuscript by Liu et al describes a new role for the E3 ubiquitin ligase Pellino1 in negatively regulating the non-canonical NF κ B pathway and so acting to suppress lupus-like conditions. The study also describes a plausible mechanism underlying these effects by showing that Pellino1 promotes ubiquitination and degradation of the NIK kinase, a key upstream regulator of the non-canonical NF κ B pathway. The manuscript is novel and innovative and adds significantly to the emerging roles of the Pellino family in immunity and disease. The study is well supported by impressive use of in vivo models in conjunction with mechanistic analysis of molecular and cellular models.

The overall conclusions are well supported by the data but there some issues that need to be addressed by the authors:

Figure 1e: The authors should discuss/speculate on the reasons why the greatest effects in Peli1-deficient cells are apparent with the IgG2a subtype.

Figure 2e: The authors should include blots of NIK expression

Figure 3b; The author's model would predict less processing of full length p100 in the WT versus KO cells and yet both cell types display the same levels of p100. The authors should comment on this discrepancy.

Figure 3f; The authors should include blots of p100 and NIK

Figure 5: Figure 5b shows strong basal ubiquitination of NIK and Figure 5d shows strong basal interaction of Peli1 and NIK. However there is no loss of basal ubiquitination of NIK in Peli1-deficient cells. The authors need to be incorporate these findings into their model , especially on the context of cIAP proteins and the BV6 data (see below).

Figure 5b & 5c: In performing ubiquitination analysis of targeted proteins (ie. NIK), the authors need to incubate lysates with denaturing buffer prior to immunoprecipitation in order to avoid NIK-interacting proteins contributing to the ubiquitination signal.

Figure 5f: More detail should be provided in the figure legend or methods sections on the identity of the E2 enzyme. Indeed some of the details in the Methods section are a little sparse.

Figure 5g and 5h. These analyses should also include blots of total levels of NIK.

Figure 5h. This study needs to include equivalent samples that were not treated with BV6 in order to compare Peli1 knockdown in the absence or presence of BV6.

Figure 6a; Immunoblots of NIK should be included.

Page 7: Lines 168-172: "Interestingly, we also found that the p52 protein levels were markedly increased in PBMCs from SLE patients with Peli1high (hereafter called SLE-PH) as compared with that from SLE patient with Peli1low mRNA expression (hereafter called SLE-PL) (Fig. 3f)..". Presumably the authors intend to indicate that SLE-PH correlates with decreased levels of p52.

Supplementary Figure 7. The authors should incorporate cIAP proteins into this diagram since their data/discussions suggests that cIAP may act via Peli1.

Reviewer #3 (Ubiquitin, NFkb)(Remarks to the Author):

In earlier papers the authors and other labs reported that Peli1 KO mice develop autoimmunity and a SLE-like phenotype and that Peli1 prevents autoimmunity by negatively regulating T cell activation. In this paper, the authors now report that Peli1 may also suppress autoimmunity by negatively regulating the non-canonical NF-kappaB pathway in B cells. The authors find that the shRNA knock-down of Peli1 also enhances activation of the non-canonical NF-kappaB pathway in a B cell line and in other cells, indicating that the effects seen in primary B cells from the Peli1 KO mice are not caused by developmental changes and are unlikely to be caused by a problem in gene targeting when the Peli1 KO mice were originally generated. Although the authors have amassed a large amount of data, I think that there are major problems with many aspects of this paper and my opinion is that it should

not be published in Nature Comm or elsewhere until these have been addressed. The major concerns are the following

The main conclusion of this manuscript is that Peli 1 negatively regulates the non-canonical NF- κ B pathway in B cells to drive autoimmunity. The critical experiments carried out in this manuscript to draw this conclusion are based on B cells isolated from Peli1 KO mice. However, as reported in earlier papers, Peli1 KO mice develop autoimmunity spontaneously (Chang et al, 2011, Nature Immunology; 28;12(10):1002-9) and it is therefore possible that the B cells isolated from these mice are already aberrant. It is clearly evident that the basal levels of Rel B and p52 in the Peli1 KO B cells are already higher (Figs 3b and 3c), suggesting that the B cells from the Peli1 KO mice are already in a more highly activated state compared to WT cells. It is also possible that hyper-activation of the non-canonical NF- κ B pathway in Peli1 KO B cells is caused by increased expression of CD40 and the BAFF receptor. The authors therefore need to demonstrate that CD40 and BAFF receptor expression is unchanged in B cells from Peli1 KO mice to validate their conclusions. In order to prove that Peli 1 deficiency in B cells causes autoimmunity through hyper-activation of the non-canonical NF- κ B pathway, the authors need to make a B cell specific Peli1 KO to really substantiate their claims. Figure 1e. The authors do not mention the age at which the mice were immunised as this can have a major effect on the production of immunoglobulins. Therefore it is not possible to judge whether the effects observed are due to the immunisation procedure or due to the spontaneous autoimmunity. For example, the IgM is already much higher at time zero. The key control experiments have not been done which would require "sham" injections without the immunogen. By the way the figure does not say what the open and closed circles are although presumably the closed circles are the WT mice. Labelling of many other figures in the manuscript is inaccurate or absent.

Figure 2 The reviewer did not think that the results presented showed any new insights into the mechanism of autoimmunity.

Figure 3g: Why do the authors show only BAFF-induced apoptotic genes and not anti-CD-40-induced gene expression? Both should have been shown.

Figure 5 The results in this figure have led the authors to conclude that Peli1 exerts its effects by catalysing the Lys48-linked ubiquitination of NIK, leading to the proteasomal degradation of NIK to switch off the non-canonical NF- κ B pathway. I found these experiments to be poorly performed, poorly controlled, extremely confusing and entirely unconvincing for the following reasons (a) Fig 5B. In this figure wild type and Peli1 KO primary B cells were incubated for 20 hours with anti-CD40 to activate the non-canonical IKK pathway and with MG132 to inhibit the proteasome and prevent the degradation of NIK. Assuming that NT means no treatment (not explained in the legend) the left hand two lanes are control cells incubated with MG132 but not stimulated with anti-CD40. NIK is then immunoprecipitated from the cell extracts and the immunoprecipitates are immunoblotted (presumably with anti-Ubiquitin antibodies). There are no molecular mass markers and it is not clear which part of the gel is being shown, but the part that is indicated shows that the ubiquitin chains are little different in the B cells from the WT and Peli1 KO mice. So if these ubiquitin chains were attached covalently to NIK, and Peli1 was making a significant contribution to the ubiquitination of NIK, there should be a big difference, but there isn't. However, there is no evidence that any of these chains are attached to NIK and they may well be attached to other proteins co-immunoprecipitating with NIK. The only way to establish what is going on is to immunoblot with an NIK antibody, which has not been done. There is no evidence that the ubiquitin chains are linked via Lys48 only and treatment with a deubiquitylase that hydrolyses Lys48-linked ubiquitin chains specifically would be needed to establish this point, but this has not been done. Another point is that MG132 inhibits caspases as potently at the proteasome and the authors will need to clarify in the Introduction the evidence that shows that NIK is really degraded by the proteasome and not degraded by caspases or another mechanism. Finally Peli1 is not active as an E3 ligase until it is phosphorylated. This is carried out by IRAK1 and/or 4 in TLR signalling but what is activating Peli in B cells as one would expect it to be inactive in unstimulated B cells? (b) In Fig 5C FLAG-tagged NIK has been over-expressed with and without HA-tagged Peli1 or a truncated form of Peli1 lacking the C-terminal RING domain, with or without HA-

tagged ubiquitin. After immunoprecipitation with anti-FLAG, the gel has been immunoblotted but I am not sure with what. The left hand side of the uppermost part of the figure is labelled anti-ubiquitin and the right hand side NIK-ubiquitin. Was the gel blotted with anti-ubiquitin, anti-HA or anti-FLAG? There is an anti-FLAG blot, which shows that the amount deubiquitinated NIK was not decreased at all by overexpression with Peli1, meaning that almost none of the NIK was converted to a ubiquitinated species in this experiment. The figure indicates that there is little or no additional ubiquitination when Peli1 was overexpressed with NIK compared to the overexpression of NIK alone. There is no evidence in this figure that any of the ubiquitin chains formed are attached to Lys48-linked ubiquitin chains. Since Pellinos are reported to catalysed the formation of Lys48-linked and Lys63-linked ubiquitin chains, an important experiment is to see whether the ubiquitin chains can be hydrolysed by deubiquitinases specific for each type of Ub chain. Moreover, it is well established that Pellino isoforms autoubiquitinate when overexpressed so the ubiquitin chains could be attached to Peli1 itself. Essential control experiments with Peli1 overexpressed in the absence of NIK overexpression have not been included. Finally, this type of experiment is flawed in many other ways and the authors are referred to a recent methodological review that spells out the problems of this type of approach (Emmerich and Cohen (2015) BBRC 466, 1-14). Figure 5D lack an essential control using pre-immune IgG. This is very important because Peli1 (47kDa) runs very close to the immunoglobulin heavy chain. In Fig 5F, FLAG NIK has been incubated with HA-Peli1, ATP, E1, E2 and ubiquitin and after IP with anti-FLAG antibodies immunblotting has been performed with either an anti-Ubiquitin or NIK antibody – again not made clear. This experiment has all the problems of the experiment in Fig 5C and a further one. The identity of the E2 is not given in the figure legend and I cannot see it in the Methods. The authors don't seem to understand how critical this is. Many RING domain proteins can form productive complexes with many E2 conjugating enzymes in vitro and in these experiments it is the E2 and not the E3 that determines the type of ubiquitin chain that is made (papers by Klevit and others). So one can get an E3 to catalyse formation of almost any type of ubiquitin chain depending on what E2 is used. So Pellinos make Lys63-linked ubiquitin chains when the E2 is Ubc13/Uev1a, but other types of Ub chains with other E2s. These experiments therefore tell us nothing about the type of ubiquitin chain that would be made in B cells when the non-canonical NF-kappaB is activated and no conclusions can be reached from such experiments. Also, the quality of the gel in Fig 5F is very poor. In summary, although the authors provide some evidence from co-IP/ immunoblotting experiments that NIK and Peli1 may interact, whether the interaction is specific or physiologically significant is unclear, and the paper fails to provide any evidence that the Peli1 E3 ligase control the non-canonical NFkappaB pathway by ubiquitylating NIK to promote its degradation. Although the Peli1 KO mice clearly develop autoimmunity and it seems that activation of the non-canonical NF-kappaB pathway is enhanced in Peli1 KO B cells, the mechanism remains unclear. RING domains are protein-protein interaction domains with functions that can be independent of their E3 ligase activity. Also, Peli1 contain a Forkhead Association (FHA) domain that binds to threonine residues in proteins and may affect their function by protein-protein interactions. The complete knock-out of Peli1 may therefore have many effects that are independent of its E3 ligase activity. Other mechanistic experiments in the paper overexpress Peli1 and the over-expression of E3 ligases can lead to abnormal ubiquitylation of proteins and erroneous conclusions being reached (discussed in Emmerich and Cohen, 2015).

Reviewer #1 (NFkB, TF regulation) (Remarks to the Author):

In this study, Liu et al identify the protein kinase NIK as a substrate of the E3 ubiquitin ligase Pellino-1. They show that Pellino-1 binds to and mediates proteasomal K48-ubiquitination-linked degradation of NIK. By controlling NIK protein levels Pellino-1 plays a critical role in preventing activation of NIK and of the non-canonical NF-κB pathway in anti-CD40- or BAFF-mediated B cell activation. A second part of the study shows that loss of Pellino-1 enhances B-cell dependent autoantibody production and the development of systemic lupus erythematoses symptoms in a mouse model. This effect is supported by a negative correlation between Pellino-1 mRNA levels in plasma cells and SLE disease activity in samples obtained from human patients. These results define a novel interesting function of Pellino-1 and relate it to an autoimmune disease for which no clear mechanism has been firmly established. The results are carefully presented and match the author's conclusions. The evidence for the direct role of Pellino-1 in regulation of NIK is strong and supported by several independent approaches, including the usage of catalytically inactive Pellino-1 mutants, overexpression experiments and in vitro ubiquitination assays. The role of the Pellino-1-NIK complex in regulation of RelB/p52-dependent genes as suggested in the summary figure S7 lacks sufficient experimental support, e.g. ChIP-PCR assays showing that RelB/p52 dimers bind to anti-CD40- or BAFF-induced target genes in a NIK- or Pellino-1-dependent manner.

Specific points:

- *Fig. 3A: There is already a weak NF-κB/DNA complex visible in nuclear extracts of unstimulated Pellino-1 ko cells. Quantification from replicates of the bandshift experiment should be shown to judge the enhancement of basal versus inducible DNA binding of NF-κB subunits in cells deficient for Pellino-1. Supershifts using anti RelB or p52 antibodies should be included to show activation of the heterodimer as proposed in Fig. S7.*

Response

Following the reviewer's comments, we have quantified the EMSA bands in the original Fig. 3a (new Fig. 4b). The results suggested that both basal and anti-CD40-induced DNA binding of NF-κB subunits are significantly enhanced in Peli1-deficient cells as compared with that in WT cells. These data are consistent with the immunoblot results that there were more RelB and p52 detected in the nucleus of Peli1-deficient cells with or without stimulation than that of WT cells (revised Fig. 4d, 4e). In addition, we have performed the supershift assay, and found that both anti-RelB and anti-p52 antibody alone shifted nearly the entire NF-κB signals (revised Fig. 4c), suggesting that anti-CD40-induced translocation of NF-κB complexes were composed primarily of RelB-p52 heterodimers in B cells as proposed in Fig. S7.

- *Fig.5: The amounts of NIK in input and pulldown samples need to be shown in the blot panels of Fig.5B and 5H. Negative controls (e.g. IgG IPs) need to be included in the IP experiments shown in Fig. 5D, E.*

Response

Following the reviewer's suggestion, we have performed the experiments of original Fig. 5b, 5h, and included the NIK blots in input and pulldown samples. The new data were presented in revised Fig. 6b, 6h. In addition, we included the negative controls by using the anti-IgG antibody to perform the co-IP experiment, and presented the data in revised Fig. 6d, 6e.

- *Fig. 6B. Likewise, the NIK blot of the input and IP samples needs to be shown to verify that comparable amounts of NIK were immunoprecipitated.*

Response

Following the reviewer's suggestion, we have performed the experiments of original Fig. 6b, and included the NIK blots panels in input and IP samples. The result suggested that comparable amounts of NIK were immunoprecipitated to test its ubiquitination status, and presented in revised Fig. 7b.

Minor points:

- *Molecular mass markers should be shown for all immunoblot panels.*

Response

We have added the molecular mass markers for all the immunoblot panels in the revised manuscripts as suggested.

- *The manuscript requires proofreading as it contains several typos and grammatical errors, e.g. in the legend of Fig. 3 (...in total lysis (lysates) of M12 cells that (were) stimulated with.....)*

Response

We apologize for these errors, and have carefully gone through the manuscript and corrected the spelling, grammatical and typo errors.

Reviewer #2 (NFkb, Pellino) (Remarks to the Author):

The manuscript by Liu et al describes a new role for the E3 ubiquitin ligase Pellino1 in negatively regulating the non-canonical NF- κ B pathway and so acting to suppress lupus-like conditions. The study also describes a plausible mechanism underlying these effects by showing that Pellino1 promotes ubiquitination and degradation of the NIK kinase, a key upstream regulator of the non-canonical NF- κ B pathway. The manuscript is novel and innovative and adds significantly to the emerging roles of the Pellino family in immunity and disease. The study is well supported by impressive use of in vivo models in conjunction with mechanistic analysis of molecular and cellular models.

The overall conclusions are well supported by the data but there some issues that need to be addressed by the authors:

- *Figure 1e: The authors should discuss/speculate on the reasons why the greatest effects in Peli1-deficient cells are apparent with the IgG2a subtype.*

Response

We thank the reviewer for this excellent point, and we have discussed the possible reason why IgG2a are the most affected Ig subtype upon T cell dependent immunization in Peli1-deficient cells in the revised manuscript. Previous studies have suggested that STAT1 and T-bet is critical for IgG2a class-switch (PNAS 2002, 99: 5545-5550; J Immunol. 2005, 175: 7419-7424), so it is reasonable to speculate that Peli1 may regulate T cell-dependent STAT1 or T-bet activation, which cooperate with noncanonical NF- κ B signaling to mediate IgG2a induction in B cells.

- *Figure 2e: The authors should include blots of NIK expression*

Response

We have included the NIK immunoblot panel in the Fig. 2e of revised manuscripts as suggested.

- *Figure 3b; The author's model would predict less processing of full length p100 in the WT versus KO cells and yet both cell types display the same levels of p100. The authors should comment on this discrepancy.*

Response

Following the reviewer's comment, we have re-run the gel to detect full length p100 protein levels in anti-CD40 or BAFF stimulated WT and KO B cells. The results suggested that there are indeed less processing of full length p100 in WT versus KO cells upon stimulation of noncanonical NF- κ B inducers, and we have included the data in the original Fig. 3b, 3c (revised Fig. 4d, 4e).

- *Figure 3f; The authors should include blots of p100 and NIK*

Response

We have included the p100 and NIK immunoblot panels in the original Fig. 3f (revised Fig. 4h) of revised manuscripts as suggested.

- *Figure 5: Figure 5b shows strong basal ubiquitination of NIK and Figure 5d shows strong basal interaction of Peli1 and NIK. However there is no loss of basal ubiquitination of NIK in Peli1-deficient cells. The authors need to be incorporate these findings into their model, especially on the context of cIAP proteins and the BV6 data (see below).*

Response

We thank the reviewer for this comment. Actually, we indeed observed a slight decrease of Lys48-linked ubiquitination of NIK both in Peli1-deficient cells and in Peli1 knockdown cells under non-stimulation condition (revised Fig. 6b, 6h). These results are in concert with the data that there is a strong basal interaction of Peli1 and NIK (revised Fig. 6d, Supplementary Fig. 6d), and increased nuclear translocation of p52 and RelB in unstimulated Peli1-deficient B cells (revised Fig. 4d, 4e). After stimulation, Peli1 deficiency led to much more dramatic reduction of NIK ubiquitination, which suppressed NIK degradation and promoted the activation of noncanonical NF- κ B signaling accordingly. We have also

incorporated these results with the BV6 data into the model presented in Supplementary Fig. 7.

- *Figure 5b & 5c: In performing ubiquitination analysis of targeted proteins (ie. NIK), the authors need to incubate lysates with denaturing buffer prior to immunoprecipitation in order to avoid NIK-interacting proteins contributing to the ubiquitination signal.*

Response

We apologize for not clearly indicating the protocol for the ubiquitination assay. Indeed, we incubated lysates with denaturing buffer prior immunoprecipitation to avoid the interference of NIK-interacting proteins for the ubiquitination analysis. In brief, the B cells were pretreated with MG132 for 1 h, and left unstimulated or stimulated with anti-CD40 for 4 h, and then were lysed with cell lysis buffer containing protease and deubiquitinase inhibitor. The cell extracts were boiled for 5 min in the presence of 1% SDS to dissociate the NIK-interacting proteins, and then were diluted with lysis buffer till the concentration of SDS was 0.1% before immunoprecipitation. NIK was then immunoprecipitated from the cell extracts and the immunoprecipitates are immunoblotted with anti-Lys48 ubiquitin antibody. We have included the detailed protocol in the Method section of the revised manuscript.

- *Figure 5f: More detail should be provided in the figure legend or methods sections on the identity of the E2 enzyme. Indeed some of the details in the Methods section are a little sparse.*

Response

We apologize for not clearly indicating the E2 enzyme (UbcH5c) used in the *in vitro* ubiquitination assay, and included this information in the revised figure legend and Method sections. In addition, we have added more details as possible as we can in the Methods section of the revised manuscript.

- *Figure 5g and 5h. These analyses should also include blots of total levels of NIK.*

Response

We have included the NIK immunoblot panels in the original Fig. 5g, 5h (revised Fig. 6g, 6h) of revised manuscripts as suggested.

- *Figure 5h. This study needs to include equivalent samples that were not treated with BV6 in order to compare Peli1 knockdown in the absence or presence of BV6.*

Response

We thank the reviewer for this excellent suggestion, since this new experiment led to a comprehensive knowledge of how Peli1 regulate noncanonical NF- κ B signaling. We found that Peli1 knockdown or BV6-induced cIAPs suppression alone could significantly inhibit Lys48-linked ubiquitination of NIK. In addition, Peli1 knockdown in the presence of BV6 further suppressed the NIK ubiquitination as compared with the effect induced by Peli1 knockdown or BV6 alone. These results confirmed our point that Peli1 serves as a direct E3 ligase of NIK, and mediates Lys48 ubiquitination and degradation of NIK independent of cIAPs. We also included these data in the revised Fig. 6h, and discussed this point revised manuscript.

- *Figure 6a; Immunoblots of NIK should be included.*

Response

We have included the NIK immunoblot panels in the original Fig. 6a (revised Fig. 7a) of revised manuscripts as suggested.

- *Page 7: Lines 168-172: "Interestingly, we also found that the p52 protein levels were markedly increased in PBMCs from SLE patients with Peli1^{high} (hereafter called SLE-PH) as compared with that from SLE patient with Peli1^{low} mRNA expression (hereafter called SLE-PL) (Fig. 3f)." Presumably the authors intend to indicate that SLE-PH correlates with decreased levels of p52.*

Response

We apologize for this error. Based on the data presented in original Fig. 3f (revised Fig. 4h), we intended to demonstrate that "we also found that the p52 protein levels were markedly increased in PBMCs from SLE patients with Peli1^{low} (hereafter called SLE-PL) as compared with that from SLE patient with Peli1^{high} mRNA expression (hereafter called SLE-PH)." We have corrected this error in the revised manuscript.

- *Supplementary Figure 7. The authors should incorporate cIAP proteins into this diagram since their data/discussions suggests that cIAP may act via Peli1.*

Response

Following the reviewer's suggestion, we have incorporated cIAPs into the diagram of revised Supplementary Fig. 7 to demonstrate how Peli1 negatively regulate noncanonical NF-κB.

Reviewer #3 (Ubiquitin, NFκb) (Remarks to the Author):

In earlier papers the authors and other labs reported that Peli1 KO mice develop autoimmunity and a SLE-like phenotype and that Peli1 prevents autoimmunity by negatively regulating T cell activation. In this paper, the authors now report that Peli1 may also suppress autoimmunity by negatively regulating the non-canonical NF-kappaB pathway in B cells. The authors find that the shRNA knock-down of Peli1 also enhances activation of the non-canonical NF-kappaB pathway in a B cell line and in other cells, indicating that the effects seen in primary B cells from the Peli1 KO mice are not caused by developmental changes and are unlikely to be caused by a problem in gene targeting when the Peli1 KO mice were originally generated. Although the authors have amassed a large amount of data, I think that there are major problems with many aspects of this paper and my opinion is that it should not be published in Nature Comm or elsewhere until these have been addressed.

The major concerns are the following

- *The main conclusion of this manuscript is that Peli 1 negatively regulate the non-canonical NF-κB pathway in B cell to drive autoimmunity. The critical experiments carried out in this manuscript to draw this conclusion are based on B cells isolated from Peli1 KO mice. However, as reported in earlier papers, Peli1 KO mice develops*

autoimmunity spontaneously (Chang et al, 2011, Nature immunology; 28;12(10):1002-9) and it is therefore possible that the B cells isolated from these mice are already aberrant. It is clearly evident that the basal levels of Rel B and p52 in the Peli1 KO B cells are already higher (Figs 3b and 3c), suggesting that the B cells from the Peli1 KO mice are already in a more highly activated state compared to WT cells. It is also possible that hyper-activation of the non-canonical NF-kappaB pathway in Peli1 KO B cells is caused by increased expression of CD40 and the BAFF receptor. The authors therefore need to demonstrate that CD40 and BAFF receptor expression is unchanged in B cell from Peli1 KO mice to validate their conclusions. In order to prove that Peli 1 deficiency in B cell causes autoimmunity through hyper activation of the non-canonical NF-kappaB pathway, the authors need to make a B cell specific Peli1 KO to really substantiate their claims.

Response

We thank the reviewer for this excellent comment. Indeed, our previous study has suggested that Peli1 deficiency lead to the spontaneously autoimmunity in old mice (> 6 months) because of the hyperactivation of Peli1 KO T cell. Nevertheless, we didn't observe any autoimmunity symptom in the young Peli1-deficient mice (5-8 week). Considering the spontaneously autoimmunity in old Peli1-deficient mice, all the B cells used in this study were collected from the mice aged around 6-7 week, and all the in vivo study were also carried out in 6-7-week-old mice. In addition, we examined the expression of CD40 and BAFF receptor on the surface of WT and Peli1-deficient B cells as suggested, and found that there is no difference of their expression between WT and Peli1-KO B cells (revised Fig. 4a), which exclude the possibility that B cells from the Peli1 KO mice are more sensitive to noncanonical NF-kB inducers (anti-CD40 and BAFF) compared to WT cells.

We also noticed that the basal levels of nuclear RelB and p52 in the Peli1 KO B cells are already slightly higher. This is because Peli1 is associated with NIK under unstimulated condition (revised Fig. 6d), and mediates the Lys48-linked ubiquitination and degradation of NIK in resting cells (revised Fig. 6b, 6h). Therefore, NIK is accumulated in Peli1-deficient B cell even without stimulation, activate downstream signaling, and finally promote the basal nuclear translocation of Rel B and p52 in Peli1-KO B cells.

Following the reviewer's comment, in order to prove that Peli1 deficiency in B cell causes autoimmunity through hyper activation of the non-canonical NF-kappaB pathway, we constructed the mixed bone marrow chimeric mice (revised Fig. 3a) by reconstituting the lethal dose irradiated B cell-deficient μ MT mice with the mixed bone marrows from μ MT mice and Peli1 WT/KO mice (μ MT : Peli1 WT/KO = 5: 1). In these chimeric mice, bone marrows from μ MT mice will provide all the genetic-competent immune cells except for B cells, and bone marrows from Peli1 WT/KO mice will provide the Peli1-competent or Peli1-deficient B cells. Since there is no B cell conditional Peli1 KO mouse available, we applied these chimeric mice to mimics the B cell specific Peli1 KO condition in vivo. By using these chimeric mice, we found that KLH immunization induced significantly increased production of serum antigen-specific antibodies in Peli1 KO/ μ MT chimeric mice as compared to WT/ μ MT chimeric mice (revised Fig. 3b). In addition, Peli1 KO/ μ MT chimeric mice developed more severe lupus-like disease than WT/ μ MT chimeric mice, as characterized by increased IgG deposition in the kidney, elevated frequencies of plasma cells and GC B cells,

and enhanced production of serum anti-nuclear antibody, and the serum levels of IgG against dsDNA, ssDNA and histone (revised Fig. 3c-3g). These results collectively confirmed B cell specific role of Peli1 in mediating the autoantibody production and lupus-like autoimmunity.

- *Figure 1e. The authors do not mention the age at which the mice were immunised as this can have a major effect on the production of immunoglobulins. Therefore it is not possible to judge whether the effects observed are due to the immunisation procedure or due to the spontaneous autoimmunity. For example, the IgM is already much higher at time zero. The key control experiments have not been done which would require “sham” injections without the immunogen. By the way the figure does not say what the open and closed circles are although presumably the closed circles are the WT mice. Labelling of many other figures in the manuscript is inaccurate or absent.*

Response

We apologize for not clearly indicating the detailed information of some experiments. Considering the spontaneously autoimmunity in old Peli1-deficient mice (> 6 months), all the B cells used in this study were collected from the mice aged around 6-7 week, and all the in vivo study were also carried out in 6-7-week-old mice, in which we didn't observe any spontaneous autoimmunity symptom. In addition, we obtained similar results that Peli1 deficiency specifically in B cells promotes the autoantibody production and lupus-like autoimmunity by using the μ MT chimeric mice (revised Fig. 3), which fully excluded the possibility of spontaneous autoimmunity induced by the hyperactivation Peli1-deficient T cell. We have specified the age of the mice used for immunization in the revised manuscript.

About the concern that IgM concentration is already higher at time zero of immunization. We reasoned that there is a strong basal interaction of Peli1 and NIK (revised Fig. 6d), which induced a slight decrease of Lys48-linked ubiquitination and degradation of NIK under non-stimulation condition (revised Fig. 6b, 6h). Accordingly, increased NIK in resting Peli1-deficient B cells promoted the activation of noncanonical NF- κ B signaling (revised Fig. 4d, 4e), leading to the IgM concentration is already higher at time zero of immunization.

We apologize for missing the labels in the original figures, and included all the labels in the revised figures.

- *Figure 2 The reviewer did not think that the results presented showed any new insights into the mechanism of autoimmunity.*

Response

We thank the reviewer for this comment. In Fig. 2, we intended to demonstrate that Peli1 deficiency lead to promoted B cell-related lupus-like autoimmunity (Fig. 2a, 2b) and increased production of autoantibodies (Fig. 2c, 2d), which may be attribute to the hyperactivation of noncanonical NF-KB signaling in B cells (Fig. 2e). Although we could not conclude the B cells specific function of Peli1 in mediating the lupus-like autoimmunity by these data only, we still could speculate that Peli1 may be involved in the regulation of B cell-mediated lupus-like autoimmunity. Moreover, the data in Fig. 2 further promoted us to apply the μ MT Peli1 WT/KO chimeric mice (revised Fig. 3) and the Rag1 mice that

transferred with Peli1 WT/KO B cells (revised Fig. 5) to confirm the specific role of Peli1 in B cells in mediating noncanonical NF- κ B signaling and lupus-like autoimmunity.

- *Figure 3g: Why do the authors show only BAFF-induced apoptotic genes and not anti-CD-40-induced gene expression? Both should have been shown.*

Response

Following the reviewer's comment, we have included the gene expression profiles induced by anti-CD40, which exhibit similar effect as BAFF, together with that induced by BAFF in the revised manuscript.

- *Figure 5 The results in this figure have led the authors to conclude that Peli1 exerts its effects by catalysing the Lys48-linked ubiquitination of NIK, leading to the proteasomal degradation of NIK to switch off the non-canonical NF-kappaB pathway. I found these experiments to be poorly performed, poorly controlled, extremely confusing and entirely unconvincing for the following reasons (a) Fig 5B. In this figure wild type and Peli1 KO primary B cells were incubated for 20 hours with anti-CD40 to activate the non-canonical IKK pathway and with MG132 to inhibit the proteasome and prevent the degradation of NIK. Assuming that NT means no treatment (not explained in the legend) the left hand two lanes are control cells incubated with MG132 but not stimulated with anti-CD40. NIK is then immunoprecipitated from the cell extracts and the immunoprecipitates are immunoblotted (presumably with anti-Ubiquitin antibodies) . There are no molecular mass markers and it is not clear which part of the gel is being shown, but the part that is indicates that the ubiquitin chains are little different in the B cells from the WT and Peli1 KO mice. So if these ubiquitin chains were attached covalently to NIK, and Peli1 was making a significant contribution to the ubiquitination of NIK, there should be a big difference, but there isnt. However, there is no evidence that any of these chains are attached to NIK and they may well be attached to other proteins co-immunoprecipitating with NIK. The only way to establish what is going on is to immunoblot with an NIK antibody, which has not been done. There is no evidence that the ubiquitin chains are linked via Lys48 only and treatment with a deubiquitylase that hydrolyses Lys48-linked ubiquitin chains specifically would be needed to establish this point, but this has not been done. Another point is that MG132 inhibits caspases as potently at the proteasome and the authors will need to clarify in the Introduction the evidence that shows that NIK is really degraded by the proteasome and not degraded by caspases or another mechanism. Finally Peli1 is not active as an E3 ligase until it is phosphorylated. This is carried out by IRAK1 and/or 4 in TLR signalling but what is activating Peli in B cells as one would expect it to be inactive in unstimulated B cells?*

Response

We apologize for not clearly indicating the related information of the original Fig. 5b. Indeed, NT represents no treatment with anti-CD40, and we have specified this information in the related figure legend. We also included all the molecular mass markers for all the immunoblot panels in the revised manuscripts.

We thank the reviewer for the comment about original Fig. 5b, and have performed new experiment as in original Fig. 5b to confirm the role of Peli1 in mediating Lys48-linked

ubiquitination of NIK. The results suggested that the Lys48-linked ubiquitination of NIK was slightly reduced in the resting Peli1-deficient B cells as compared with WT cells, whereas anti-CD40 stimulation dramatically impaired the Lys48-linked ubiquitination of NIK in Peli1-deficient B cells. These results suggested that Peli1 may be not fully activated to function as an E3 ligase for NIK ubiquitination until upon the stimulation of noncanonical NF- κ B inducers. Previous study has identified TBK1 as a critical protein kinase responsible for Peli1 phosphorylation and activation (Biochem J. 2011, 434: 537-48.), and we also suggested that TBK1 is activated in B cells upon the stimulation of noncanonical inducers (including anti-CD40 and BAFF) (Nat Immunol. 2012, 13: 1101-1109). Thus, it is reasonable to speculate that TBK1 is the key protein kinase that responsible for the phosphorylation and activation of Peli1, which is then activated to mediate the Lys48-linked ubiquitination and degradation of NIK in noncanonical NF- κ B signaling. We have included the new data in the revised Fig. 6b, and discussed this issue in the Discussion section of the revised manuscript.

About the reviewer's concern of the protocol used in our study to detect NIK ubiquitination, we have specified the detailed protocol for these experiments in the Method section of the revised manuscript. In brief, the B cells were pretreated with MG132 for 1 h, and left unstimulated or stimulated with anti-CD40 for 4 h, and then were lysed with cell lysis buffer containing protease and deubiquitinase inhibitor. The cell extracts were boiled for 5 min in the presence of 1% SDS to dissociate the NIK-interacting proteins, and then were diluted with lysis buffer till the concentration of SDS was 0.1% before immunoprecipitation. NIK was then immunoprecipitated from the cell extracts and the immunoprecipitates are immunoblotted with anti-ubiquitin or anti-Lys48 ubiquitin antibody. Since there is no good anti-NIK antibody available for the direct detection of its ubiquitination status, we applied this IP NIK, and then anti-ub immunoblot protocol, which is also the most commonly used protocol, to examine Peli1-mediated NIK ubiquitination.

In addition, we confirmed that overexpression of the non-specific deubiquitinases USP2 almost hydrolyzed all the ubiquitin chains of NIK that induced by full-length Peli1, suggesting Peli1 indeed mediate NIK ubiquitination but not the other modifications, and included these data in the revised Fig. 6c. We also confirmed that Peli1 catalyzed the formation of Lys48-linked ubiquitin chains of NIK in the revised Fig. 6b, 6f, 6h, 7b, Supplementary Fig. 6a, 6c.

About the reviewer's concern that MG132 may potentially inhibits caspases, we modified the demonstration in the Introduction section to "...thereby promoting cIAP-mediated Lys48-linked NIK polyubiquitination and degradation", since there is no direct evidence that shows NIK is really degraded by the proteasome and not degraded by caspases or another mechanism as suggested.

- *(b) In Fig 5C FLAG-tagged NIK has been over-expressed with and without HA-tagged Peli1 or a truncated form of Pellino lacking the C-terminal RING domain, with or without HA-tagged ubiquitin. After immunoprecipitation with anti-FLAG, the gel has been immunoblotted but I am not sure with what. The left hand side of the uppermost part of the figure is labelled anti-ubiquitin and the right hand side NIK-ubiquitin. Was the gel blotted with anti-ubiquitin, anti-HA or anti-FLAG? There is an anti-FLAG blot, which shows that the amount deubiquitinated NIK was not decreased at all by overexpression*

with Peli1, meaning that almost none of the NIK was converted to a ubiquitinated species in this experiment. The figure indicates that there is little or no additional ubiquitination when Peli1 was overexpressed with NIK compared to the overexpression of NIK alone. There is no evidence in this figure that any of the ubiquitin chains formed are attached to Lys48-linked ubiquitin chains. Since Pellinos are reported to catalyzed the formation of Lys48-linked and Lys63-linked ubiquitin chains, an important experiment is to see whether the ubiquitin chains can be hydrolysed by deubiquitinases specific for each type of Ub chain.

Response

Following the reviewer's suggestion, we have performed additional experiment that including proper controls to demonstrate the role of Peli1 in mediating NIK ubiquitination, and included the data in the revised Fig. 6c. The results showed the by using the anti-Flag antibody to pulldown target protein, the ubiquitination was detected only in the lanes that Flag-NIK is overexpressed with or without HA-ubiquitin, whereas no ubiquitination was detected in the lane of overexpression of HA-Peli1 plus HA-ubiquitin, suggesting only Flag-NIK is pulled down by using anti-Flag antibody and the specificity of NIK ubiquitination were detected after pulldown of Flag-NIK in this experiment. In addition, full-length Peli1 but not its RING domain deletion mutant (Peli1 Δ C, loss of E3 ligase function) markedly enhanced NIK ubiquitination, and overexpression of the non-specific deubiquitinase USP2 almost hydrolyzed all the ubiquitin chains of NIK that induced by full-length Peli1, suggesting Peli1 indeed mediate NIK ubiquitination but not the other modifications. The antibody we used to detect NIK ubiquitination in this experiment was the anti-ubiquitin antibody, and labeled at the left-hand side of the uppermost part of the revised Fig. 6c. These data collectively established the critical role of Peli1 in mediating the ubiquitination of NIK. We also confirmed that Peli1 catalyzed the formation of Lys48-linked ubiquitin chains of NIK in the revised Fig. 6b, 6f, 6h, 7b, Supplementary Fig. 6a, 6c.

- *Moreover, it is well established that Pellino isoforms autoubiquitinate when overexpressed so the ubiquitin chains could be attached to Peli1 itself. Essential control experiments with Peli1 overexpressed in the absence of NIK overexpression have not been included. Finally, this type of experiment is flawed in many other ways and the authors are referred to a recent methodological review that spells out the problems of this type of approach (Emmerich and Cohen (2015) BBRC 466, 1-14).*

Response

To address the reviewer's concern, we have performed the experiment to re-examine the NIK ubiquitination by adding proper controls. As shown in the revised Fig. 6c, we could not detect the ubiquitination in the lane that overexpressed Peli1 and ubiquitin after pulldown of Flag-NIK (anti-Flag) by using the anti-ubiquitin antibody, and the NIK ubiquitination was only detected in the presence of NIK overexpression.

- *Figure 5D lack an essential control using pre-immune IgG. This is very important because Peli1 (47kDa) runs very close to the immunoglobulin heavy chain.*

Response

Following the reviewer's suggestion, we included the negative controls by using the anti-IgG antibody to perform the co-IP experiment, and presented the data in revised Fig. 6d, 6e. In addition, considering the molecular mass of Peli1, the co-IP antibody used for detecting Peli1 is anti-Peli1-HRP, which is directly conjugated with HRP, and eliminate the interference of immunoglobulin heavy chain during immunoblot analysis. We also included the Peli1 antibody information in the revised manuscript.

- *In Fig 5F, FLAG NIK has been incubated with HA-Peli1, ATP, E1, E2 and ubiquitin and after IP with anti-FLAG antibodies immunoblotting has been performed with either an anti-Ubiquitin or NIK antibody – again not made clear. This experiment has all the problems of the experiment in Fig 5C and a further one. The identity of the E2 is not given in the figure legend and I cannot see it in the Methods. The authors don't seem to understand how critical this is. Many RING domain proteins can form productive complexes with many E2 conjugating enzymes in vitro and in these experiments it is the E2 and not the E3 that determines the type of ubiquitin chain that is made (papers by Klevit and others). So one can get an E3 to catalyse formation of almost any type of ubiquitin chain depending on what E2 is used. So Pellinos make Lys63-linked ubiquitin chains when the E2 is Ubc13/Uev1a, but other types of Ub chains with other E2s. These experiments therefore tell us nothing about the type of ubiquitin chain that would be made in B cells when the non-canonical NF-kappaB is activated and no conclusions can be reached from such experiments. Also, the quality of the gel in Fig 5F is very poor.*

Response

We apologize for not clearly indicating the E2 enzyme used in the in vitro ubiquitination assay. The E2 used for in vitro ubiquitination assay is UbcH5c, which has been reported to be capable of mediating the Lys48-linked ubiquitination, and we have included this information in the revised figure, figure legend and Method sections. The antibody we used to detect in vitro ubiquitination of NIK is anti-Lys48 ubiquitin antibody, which only detect Lys48-linked ubiquitin chain for immunoblot analysis. We have indicated the antibody information in the revised Fig. 6f. Therefore, these results suggested that the E3 ligase Peli1 indeed mediate Lys48-linked ubiquitination of NIK.

About the reviewer's concern raised by original Fig. 5c, we have performed additional experiment that including proper controls, and the detailed responses have been demonstrated below the abovementioned comment about the original Fig. 5c.

- *In summary, although the authors provide some evidence from co-IP/ immunoblotting experiments that NIK and Peli1 may interact, whether the interaction is specific or physiologically significant is unclear, and the paper fails to provide any evidence that the Peli1 E3 ligase control the non-canonical NFkappaB pathway by ubiquitylating NIK to promote its degradation. Although the Peli1 KO mice clearly develop autoimmunity and it seems that activation of the non-canonical NF-kappaB pathway is enhanced in Peli1 KO B cells, the mechanism remains unclear. RING domains are protein-protein interaction domains with functions that can be independent of their E3 ligase activity. Also, Peli1 contain a Forkhead Association (FHA) domain that binds to threonine residues in proteins and may affect their function by protein-protein interactions. The*

complete knock-out of Peli1 may therefore have many effects that are independent of its E3 ligase activity. Other mechanistic experiments in the paper overexpress Peli1 and the over-expression of E3 ligases can lead to abnormal ubiquitylation of proteins and erroneous conclusions being reached (discussed in Emmerich and Cohen, 2015).

Response

We thank the reviewer for this comment, and respond the reviewer's concerns as follows.

For the concern about Peli1-NIK interaction, we observed strong associations between Peli1 and NIK when NIK was pulled down both in primary B cells with or without stimulation, and in 293T cells that overexpressed with these two proteins. In addition, there is no non-specific binding detected by using the anti-IgG antibody to pulldown, suggesting the interaction between Peli1 and NIK is specific (revised Fig. 6d, 6e).

For the concern about Peli1 controls noncanonical NF- κ B signaling through NIK ubiquitination, we demonstrated in the revised Fig. 4 that Peli1 deficiency or knockdown leads to promoted noncanonical NF- κ B activation, and found in the revised Fig. 6 that Peli1 serves as a direct E3 ligase of NIK, and mediates Lys48 ubiquitination and degradation of NIK. Previous studies have demonstrated that NIK is the master kinase responsible for the activation of noncanonical NF- κ B pathway (Nat Immunol. 2008, 9:1364–1370; Nat Immunol. 2008, 9:1371–1378; Immunol Rev. 2012, 246:125–140), so it is reasonable to assume that the promoted noncanonical NF- κ B activation in Peli1-deficient cells is attribute to the accumulation of NIK, which results from the lost control of NIK ubiquitination and degradation upon the deletion of Peli1. In addition, as shown in the revised Fig. 7, reconstitution of full-length Peli1 could inhibit the noncanonical NF- κ B activation and promote the Lys48-linked ubiquitination of NIK accordingly in Peli1 knockdown cells, whereas reconstitution of RING domain deletion Peli1 (without E3 ligase activity) failed promote NIK ubiquitination and inhibit noncanonical NF- κ B activation. These results further confirmed that Peli1 indeed controls noncanonical NF- κ B signaling through NIK ubiquitination.

For the concern about the RING domain of Peli1, we agreed with the reviewer that RING domains are protein-protein interaction domains with functions that can be independent of their E3 ligase activity, and the complete knock-out of Peli1 may therefore have effects that are independent of its E3 ligase activity. In this case, Peli1 knockin mice that only deficient for its E3 ligase activity is needed for the future studies, and we have discussed this issue in the revised manuscript.

For the concern about Peli1 overexpression, we agreed with the reviewer that the over-expression of E3 ligases can lead to abnormal ubiquitylation of proteins. Nevertheless, this is the most commonly used method to examine the effect of the ubiquitination of a target protein induced by a specific E3 ligase. Considering the possible artificial effect in this overexpression system, we not only found that Peli1 mediate NIK ubiquitination in 293T overexpression system (revised Fig. 6c), but also observed that suppressed endogenous NIK ubiquitination in Peli1 deficient or knockdown cells (revised Fig. 6b, 6h). These data collectively confirmed that Peli1 indeed mediate the regulation of NIK ubiquitination.

REVIEWERS' COMMENTS:

Reviewer #1 (Remarks to the Author):

The authors have appropriately addressed my previous concerns.

Reviewer #2 (Remarks to the Author):

The authors have adequately addressed all concerns raised by this reviewer by inclusion of additional data and/or with editorial changes. The study represents a further and valuable insight into the physiological role of Peli1 and also provides understanding of its mechanism of action. It represents a significant advance in the field.

Reviewer #4 (Remarks to the Author):

The revised manuscript describes a new, negative regulatory function of Peli1 in non conical NF- κ B activation and SLE disease model. The overall conclusion is that Peli1 functions as an E3 ligase to ubiquitinate NIK via K48-linked chain and promote its degradation. The authors tested not only biochemically, but also into the "reconstituted" mouse model (or adoptive transfer) and SLE mouse model. Further data derived from SLE patients also support the negative regulation of NF κ B activation by Peli1. These findings are significant and will certainly advance our understanding in NF κ B regulation and its related diseases, such as SLE.

Overall, the authors have performed a number of important experiments to address previous concerns and criticisms. These results further solidified their conclusions.

Previous reviewer 3 had significant number of questions concerning the specificity of ubiquitin chain linkage of NIK by Peli1. This is an important scientific question. In the revised version, the authors now shown that an antibody specific for the K48-linked ubiquitin chains reacted well with the ubiquitinated NIK from in cell and in vitro system. This is an important piece of evidence. Additional experiments further supporting their conclusion include to sequence the ubiquitinated NIK by mass spectrometry to show the K48-linkage, and that expression of exogenous K48R mutant of ubiquitin can block NIK ubiquitination. These experiments are technically feasible and will further support the authors' conclusion.

Reviewer #1 (Remarks to the Author):

- *The authors have appropriately addressed my previous concerns.*

Response

We thank the reviewer for the positive comments.

Reviewer #2 (Remarks to the Author):

- *The authors have adequately addressed all concerns raised by this reviewer by inclusion of additional data and/or with editorial changes. The study represents a further and valuable insight into the physiological role of Peli1 and also provides understanding of its mechanism of action. It represents a significant advance in the field.*

Response

We thank the reviewer for the positive comments.

Reviewer #4 (Remarks to the Author):

- *The revised manuscript describes a new, negative regulatory function of Peli1 in non canonical NF- κ B activation and SLE disease model. The overall conclusion is that Peli1 functions as an E3 ligase to ubiquitinate NIK via K48-linked chain and promote its degradation. The authors tested not only biochemically, but also into the "reconstituted" mouse model (or adoptive transfer) and SLE mouse model. Further data derived from SLE patients also support the negative regulation of NF κ B activation by Peli1. These findings are significant and will certainly advance our understanding in NF κ B regulation and its related diseases, such as SLE.
Overall, the authors have performed a number of important experiments to address previous concerns and criticisms. These results further solidified their conclusions.*

Response

We thank the reviewer for the positive comments.

- *Previous reviewer 3 had significant number of questions concerning the specificity of ubiquitin chain linkage of NIK by Peli1. This is an important scientific question. In the revised version, the authors now shown that an antibody specific for the K48-linked ubiquitin chains reacted well with the ubiquitinated NIK from in cell and in vitro system. This is an important piece of evidence. Additional experiments further supporting their conclusion include to sequence the ubiquitinated NIK by mass spectrometry to show the K48-linkage, and that expression of exogenous K48R mutant of ubiquitin can block NIK ubiquitination. These experiments are technically feasible and will further support the authors' conclusion.*

Response

We thank the reviewer for the positive comments and constructive suggestions. For the concern about Peli1-mediated K48 ubiquitination, we actually have performed the experiment and confirmed that Peli1 fails to promote the NIK ubiquitination by using K48R mutant of ubiquitin, and included the result in the revised Supplementary Fig. 5b, which suggested that Peli1 indeed mediates the formation of K48-linked poly-ubiquitination chains of NIK.